# How to Retrain Online Models Optimally with Few Updates

## Abstract

Retraining is the primary mechanism by which AI models can update their internal parameters in response to evolving environments, yet it is also one of the most costly operations. This raises a fundamental question: *how often must a model be retrained to achieve optimal performance over time?* We address this problem in the framework of *online learning*, beginning with the classical but foundational case of i.i.d. realizable data. We show that retraining at every step is unnecessary: in most cases, only $O(\log T)$ updates suffice to achieve near-optimal risk, where $T$ is the number of steps. Furthermore, when the *learning curve* decays as $1/t^\alpha$ with $\alpha < 1$, as few as $O(\log \log T)$ updates are enough. We design algorithms that achieve these guarantees, including adaptive methods that remain optimal when $\alpha$ is unknown, and extend our analysis to piecewise-stationary settings with *distribution shifts*. We also establish sharp impossibility results, proving that no universal algorithm exists without prior knowledge of the learning curve. Together, these results provide the first precise characterization of optimal retraining frequency, bridging foundational theory with practical strategies for scalable AI.

## 1 Introduction

Current AI systems have achieved remarkable success across a wide range of applications, enabled by training on large volumes of data that yield strong statistical concentration and effective representations. As these models are deployed over extended periods, they inevitably encounter a continual stream of new data. Incorporating such data can improve predictive accuracy, but doing so typically requires *retraining*, which updates the model's internal parameters. Retraining, however, is among the most costly operations in modern AI—whether in terms of computational resources or stability. This raises a fundamental question: *as data accumulate, how often should retraining occur to maintain optimal predictive performance?*

We address this problem through the lens of *online learning* (Shalev-Shwartz & Ben-David, 2014), a setting in which data arrive sequentially over rounds $t = 1, 2, \ldots, T$ and the learner must make predictions in each round while updating its hypothesis based on feedback. Formally, online learning can be phrased as a game between a learner and nature. At each round $t$, the learner selects a predictor $\hat{h}_t$ based on past data; upon receiving a new instance $X_t$, it outputs a prediction $\hat{Y}_t := \hat{h}_t(X_t)$. Nature then reveals the true label $Y_t$, and the learner incurs a loss $\ell(\hat{Y}_t, Y_t)$ for some predefined loss function $\ell$. The performance of the learner over $T$ rounds is measured by its cumulative risk

$$\mathsf{Risk}_T(\{(X_t, Y_t)\}_{t=1}^T) \;=\; \sum_{t=1}^T \ell\big(\hat{h}_t(X_t), Y_t\big).$$

While natural, this paradigm implicitly assumes that *retraining* (i.e., updating $\hat{h}_t$) at every step $t$ is essential for maintaining optimal performance. A fundamental question that has remained largely unexplored is whether this assumption is truly necessary: can comparable accuracy be achieved with dramatically fewer updates? In this work, we show that the answer is affirmative, by providing the first precise characterization of the trade-off between retraining frequency and predictive risk. Our results reveal that exponentially fewer updates are sufficient in many regimes, and we design algorithms that achieve these guarantees with sharp theoretical bounds.

More precisely, we assume access to an *oracle* $\mathcal{O}$ that, given any dataset $\{(X_i, Y_i)\}_{i=1}^m$, returns a predictor $\hat{h} \in \mathcal{H}$ from a hypothesis class $\mathcal{H}$. Retraining corresponds to invoking the oracle. We allow at most $S$ such retraining steps (i.e., oracle calls), which occur at chosen time points: $0 = \tau_0 < \tau_1 < \cdots < \tau_S < T$. At each update time $\tau_s$, the oracle produces a predictor $\hat{h}_s$ based on the data observed so far. This predictor is then deployed throughout the $s$-th epoch, i.e., $\forall t \in [\tau_s, \tau_{s+1})$ we predict $\hat{Y}_t = \hat{h}_s(X_t)$.

Given a retraining budget $S$, our goal is to design the update schedule $\tau_1, \ldots, \tau_S$ so as to minimize the cumulative risk $\mathsf{Risk}_{S,T}$ in a specified learning environment (see Section 2 for definition). We also focus on the *minimal* budget $S$ for which $\mathsf{Risk}_{S,T}$ asymptotically *matches* the risk $\mathsf{Risk}_{T,T}$ that would be attained if retraining were performed at *every* step (i.e., $S = T$).

**Our Contributions.** We start with the classical i.i.d. realizable setting, which already captures the fundamental trade-off between the retraining budget and the predictive risk. In this case, the data $\{(X_t, Y_t)\}_{t=1}^T$ are assumed to be drawn i.i.d. from an *unknown* distribution $\mu$. Our analysis relies on the concept of *learning curve* $\bar{L}(t)$ (see Eq. (4) for a precise definition), defined as:

$$\bar{L}(t) := \mathbb{E}_{\mathcal{D}_t} \, \mathbb{E}_{(X,Y)\sim\mu} \Big[ \ell\big(\hat{h}_{\mathcal{D}_t}(X), Y\big) \Big],$$

where $\mathcal{D}_t := \{(X_i, Y_i)\}_{i=1}^t \overset{\text{i.i.d.}}{\sim} \mu$, $\hat{h}_{\mathcal{D}_t} := \mathcal{O}(\mathcal{D}_t)$ for some oracle $\mathcal{O}$ (e.g., an ERM rule), and $(X, Y)$ denotes a fresh sample from the same distribution $\mu$.

Our first main result, Theorem 1, establishes that only $\log T$ updates suffice to achieve a risk within a constant factor of the optimal *full-update* risk, provided that the learning curve $\bar{L}(t)$ is *non-increasing*, regardless of its exact shape. The proof is reminiscent of the classical "doubling trick," however, our result is more general that applies to *any* non-increasing learning curve, without assuming a specific functional form.

We then establish the *precise* optimal risk when additional properties of the learning curve are satisfied. We show in Theorem 2 that if $\bar{L}(t) = \tilde{O}(1/t)$, e.g., when $\mathcal{H} \subset \{0, 1\}^{\mathcal{X}}$ has finite VC-dimension and the oracle $\mathcal{O}$ is ERM, the optimal updates are scheduled at $\tau_i = T^{i/(S+1)}$ for $1 \leq i \leq S$, leading to the expected risk $\mathbb{E}[\mathsf{Risk}_{S,T}] \leq \tilde{O}((S+1) \cdot T^{\frac{1}{S+1}})$. For instance, when $S = 1$, this yields an update at $\tau_1 = \sqrt{T}$ with cumulative risk of order $\sqrt{T}$.

When the learning curve follows *power law* $\bar{L}(t) = \tilde{\Theta}(t^{-\alpha})$ for some $\alpha < 1$, Theorem 3 provides an update schedule that yields $\mathbb{E}[\mathsf{Risk}_{S,T}] \leq \tilde{O}(T^{\frac{1-\alpha}{1-\alpha^{S+1}}})$. Perhaps surprisingly, this implies that only $S = O(\log \log T)$ updates are sufficient to achieve the optimal *full-update* risk. We further design an update rule (Algorithm 1) that leverages the *empirical losses* incurred by the predictors, and attains the optimal risk (Theorem 5) even when the exponent $\alpha$ is *unknown*.

Beyond the i.i.d. setting, our results also extend to *piecewise-i.i.d.* data distributions. For this regime, we propose an efficient update procedure (Algorithm 6) that achieves near-optimal risk under distribution shifts, as shown in Theorem 6. Finally, we show in Theorem 7 that, without any prior information about the learning environment, no universal algorithm can achieve competitive optimality (even up to polylogarithmic factors) relative to the case where the environment is known. Finally, we provide empirical validation on real-world dataset and models in Appendix B.

**Technique Overview.** Before delving into the technical details, we illustrate our main ideas in Figure 1. The full-update risk $\mathsf{Risk}_{T,T}$ can be interpreted as the area under the learning curve $\bar{L}(t)$, while the restricted-update risk $\mathsf{Risk}_{S,T}$ corresponds to the area under a step function defined by updates at times $\tau_1, \ldots, \tau_S$. As the number of updates grows, the approximation error—the shaded region in the figure—becomes asymptotically negligible compared to the total risk. The central question is: how many updates are actually necessary? Theorem 1 shows that $\log T$ updates suffice under the mild assumption that $\bar{L}(t)$ is *non-increasing*. With stronger structural assumptions on $\bar{L}(t)$, even sharper guarantees follow, as established in Theorems 2–6.

**Related Work.** Online learning has been extensively studied in the literature, where updates are assumed at every time step (Rakhlin & Sridharan, 2015; Shalev-Shwartz & Ben-David, 2014; Foster et al., 2018; Wu et al., 2022; 2025). In contrast, literature that explicitly investigates the cost of

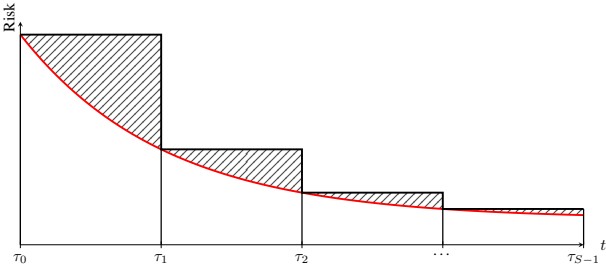

Figure 1: Learning curve $\bar{L}(t)$ (red) approximated by a step function (gray) induced by update rule, with the approximation error shown as shaded areas.

retraining in online learning is scarce. Notable exceptions are Devulapalli & Hanneke (2024); Devulapalli et al. (2025), which consider a *continuous-time* online model that queries samples only at selected time steps and updates the model at those query times. However, their work does not treat the update schedule as a *design* objective and relies exclusively on the queried (sampled) *adversarial* examples, which can be overly pessimistic for practical scenarios. Recent efforts propose decision rules for triggering retraining based on uncertainty or cost considerations (Regol et al., 2025; Zanotti, 2025). These results provide useful heuristics and case studies, but they do not characterize the minimal update budget needed to match the predictive performance of continuous retraining. Other related areas include continual and incremental learning (He et al., 2020), as well as learning under nonstationarity and dynamic regret (Block et al., 2022; Haghtalab et al., 2020; Hazan & Seshadhri, 2009; Wu et al., 2023). Although these efforts study adaptation under distributional shift, their objectives differ from ours, which focus on the statistical limits of retraining under a fixed update budget. Finally, empirical studies of scaling laws suggest that learning curves often follow power-law behavior (Kaplan et al., 2020; Hestness et al., 2017; Hoffmann et al., 2022), which motivates the modeling assumptions adopted in this work.

## 2 PROBLEM FORMULATION

Let $\mathcal{X}$ be the context space, $\mathcal{Y}$ the label space, and $\mathcal{H} \subset \mathcal{Y}^{\mathcal{X}}$ a hypothesis class. A *training oracle* (or learning rule) is a function $\mathcal{O} : (\mathcal{X} \times \mathcal{Y})^* \to \mathcal{H}$, which takes as input a sequence of context-label pairs and returns a predictor in $\mathcal{H}$. We consider the following online learning scenario with a limited update (or retraining) budget of $S$:

---

**Protocol 1** Selective Update Online Learning Protocol

---

1: Select initial predictor $\hat{h}_0 \in \mathcal{H}$ and set update index $s = 0$
2: **for** $t = 1, \ldots, T$ **do**
3:     Use predictor $\hat{h}_s$ to make a prediction on $X_t$
4:     Observe true label $Y_t$
5:     Use update rule $\Phi$ with $\{(X_i, Y_i)\}_{i=1}^t$ and predictor $\hat{h}_s$ to determine whether to update
6:     **if** $\Phi$ triggers an update and $s < S$ **then**
7:         Select data subset $\mathcal{D}_{s+1} \subset \{(X_i, Y_i)\}_{i=1}^t$ and update predictor $\hat{h}_{s+1} = \mathcal{O}(\mathcal{D}_{s+1})$
8:         Set $s \leftarrow s + 1$
9:     **end if**
10: **end for**

---

We denote $\tau_1, \ldots, \tau_S$ as the update time steps, and define $s(t) := \max\{s \in \{1, \ldots, S\} : \tau_s \leq t\}$ as the index of the most recent update at or before time $t \in \{1, \cdots, T\}$. Our goal is to design an update rule $\Phi$ that, given access to a retraining oracle $\mathcal{O}$ and a fixed update budget $S$, minimizes the total cumulative risk incurred by the learner over a time horizon of $T$ prediction rounds. Specifically, the empirical cumulative risk (or total loss) incurred by a given *update rule* $\Phi$ (which at time $t$ can take

value 1, if update happens at this time and 0 otherwise) is defined as:

$$\mathsf{Risk}_{S,T}(\Phi, \{(X_t, Y_t)\}_{t=1}^T) := \sum_{t=1}^T \ell(\hat{h}_{s(t)}(X_t), Y_t) = \sum_{s=0}^S \sum_{t=\tau_s+1}^{\tau_{s+1}} \ell(\hat{h}_s(X_t), Y_t), \quad (1)$$

where $\ell : \mathcal{Y} \times \mathcal{Y} \to \mathbb{R}_{\geq 0}$ is a loss function and we assume $\tau_0 = 0$ and $\tau_{S+1} = T$.

In many practical settings, retraining a model is significantly more expensive than evaluating it on new data. Therefore, it is desirable to limit the number of retraining events while allowing the learner to freely test the current predictor on incoming examples. This motivates a class of update rules that rely solely on observable predictive performance.

**Risk-driven update.** An update rule $\Phi$ is said to be *risk-driven* if, at any time step $t$, it only treats the current predictor $\hat{h}_{s(t)}$ as a black box and bases its update decision solely on the *observed losses* $\{\ell(\hat{h}_{s(t)}(X_i), Y_i)\}_{i=1}^t$ over the available data. That is, $\Phi$ must determine whether to update exclusively based on the predictive performance of the current predictor on the historical data.

This paper focuses primarily on designing *risk-driven* update rules that achieve optimal cumulative risk. To proceed, we assume that the data is generated by an unknown *random process* belonging to a known class P. We refer to the tuple $(\mathsf{P}, \mathcal{H}, \mathcal{O})$ as a *learning environment*.

For any given learning environment, we first define the expected risk, for $\boldsymbol{\mu} \in \mathsf{P}$, as follows:

$$\mathsf{Risk}_{S,T}(\Phi, \boldsymbol{\mu}) := \mathbb{E}_{\{(X_t, Y_t)\}_{t=1}^T \sim \boldsymbol{\mu}} \left[ \mathsf{Risk}(\Phi, \{(X_t, Y_t)\}_{t=1}^T) \right], \quad (2)$$

Our goal is to characterize the following minimax risk:

$$\mathsf{Risk}_{S,T}(\mathsf{P}) := \inf_{\Phi} \sup_{\boldsymbol{\mu} \in \mathsf{P}} \mathbb{E} \left[ \mathsf{Risk}(\Phi, \{(X_t, Y_t)\}_{t=1}^T) \right], \quad (3)$$

where the infimum is taken over all *risk-driven* update rules $\Phi$ that satisfy the update budget constraint $S$, and the expectation is over the randomness of the data sequence $\{(X_t, Y_t)\}_{t=1}^T \sim \boldsymbol{\mu}$ as well as any internal randomness of $\Phi$ and the retraining oracle $\mathcal{O}$.

**Realizable *i.i.d.* Learning Environment.** In this setting, the data points $\{(X_t, Y_t)\}_{t=1}^T$ are drawn independently and identically from an unknown distribution $\mu$ over $\mathcal{X} \times \mathcal{Y}$. We assume there exists an unknown hypothesis $h \in \mathcal{H}$ such that $h(X) = Y$ for $(X, Y) \sim \mu$ almost surely (i.e., $\mu$ is realizable w.r.t. $\mathcal{H}$). The hypothesis class $\mathcal{H}$ is arbitrary, and the oracle $\mathcal{O}$ is assumed to return an empirical risk minimizer (ERM) over a given dataset. Let $\mathcal{D}_t := \{(X_i, Y_i)\}_{i=1}^t \overset{i.i.d.}{\sim} \mu$, and define $\hat{h}_t = \mathcal{O}(\mathcal{D}_t)$. We define the learning curve $L(t)$ and the expected learning curve $\bar{L}(t)$ as follows

$$L(t) = \mathbb{E}_{(X,Y)\sim\mu}[\ell(\hat{h}_t(X), Y)] \quad \text{and} \quad \bar{L}(t) := \mathbb{E}_{\mathcal{D}_t} \mathbb{E}_{(X,Y)\sim\mu} \left[ \ell(\hat{h}_t(X), Y) \right]. \quad (4)$$

where $(X, Y)$ is a fresh sample independent of $\mathcal{D}_t$. It is known that for a hypothesis class $\mathcal{H}$ with finite VC-dimension and for the 0-1 loss, the learning curve satisfies, for any $\mu \in \mathsf{P}$

$$\bar{L}(t) \leq O\left( \mathsf{VC}(\mathcal{H}) t^{-1} \log t \right) = \tilde{O}\left( t^{-1} \right), \quad (5)$$

where $\tilde{O}$ hides poly-logarithmic factors in $t$. It is also known that the learning curve behaves like $L(t) = O\left( \frac{\mathsf{VC}(\mathcal{H}) \log t + \log(1/\delta)}{t} \right)$ with probability at least $1 - \delta$ (Shalev-Shwartz & Ben-David, 2014).

In the case of real-valued hypothesis classes, when the loss function is Lipschitz (e.g., absolute loss) and the fat-shattering dimension scales as $\gamma^{-p}$ for some $p > 0$ (for example, for $\mathcal{H} = \{h_{\mathbf{w}}(\mathbf{x}) := \langle \mathbf{w}, \mathbf{x} \rangle : \|\mathbf{w}\|_p \leq 1, \|\mathbf{x}\|_{\frac{p}{p-1}} \leq 1\}$), it can be shown (see (Attias et al., 2023, Theorem 2) and the discussion that follows) that, for any $\mu \in \mathsf{P}$,

$$\bar{L}(t) \leq \tilde{O}\left( t^{-\frac{1}{p+1}} \right). \quad (6)$$

We refer to learning curves of the form $\bar{L}(t) = \tilde{O}(t^{-\alpha})$ with $\alpha < 1$ as exhibiting *power-law* scaling, a phenomenon that is also observed empirically in many practical settings (Kaplan et al., 2020; Hestness et al., 2017; Hoffmann et al., 2022). A concrete example where the learning curve provably follows a power law is provided in Appendix C.

Beyond i.i.d. processes, we also consider more general data processes, such as:

**Piecewise-Stationary Environment.** Data are generated independently according to a sequence of distributions $\{\mu_t\}_{t=1}^T$, where $\mu_t$ remains fixed within a contiguous segment but may *change* between segments. Here, the class P consists of all such piecewise-stationary processes with at most $K$ changes. This setting models scenarios with *concept drift* or *distribution shift*.

## 3 MAIN RESULTS

In this section, we present our main theoretical and algorithmic results, focusing primarily on the i.i.d. setting, with extensions to the piecewise-stationary case in Section 3.2.1. Empirical validation is provided in Appendix B. Throughout, we assume that the loss function $\ell(\cdot, \cdot)$ is bounded.

### 3.1 FIXED-DESIGN SCHEDULE

In this section, we present several update rules that rely solely on the *shape* of the learning curve, without using the predictive performance of the resulting predictors.

We begin with a general result showing that only $\log T$ updates are sufficient to be *competitively optimal* when the learning curve $\bar{L}(t)$ is *non-increasing*. The main idea is illustrated in Figure 1. Recall that $\mathsf{Risk}_{T,T}$ corresponds to the area under the learning curve $\bar{L}(t)$, while the risk $\mathsf{Risk}_{S,T}$ of the update schedule at $\tau_1, \cdots, \tau_S$ can be viewed as the area under the step function.

**Theorem 1.** *Consider any realizable* i.i.d. *learning environment* $(\mathsf{P}, \mathcal{H}, \mathcal{O})$. *If the update times are chosen as* $\tau_i = 2^{i-1}$ *for* $i \in \{1, \ldots, \log T\}$, *then, with* $S = \log T$, *we have*

$$\mathsf{Risk}_{S,T}(\mathsf{P}) \leq 2\,\mathsf{Risk}_{T,T}(\mathsf{P}), \tag{7}$$

*provided that the learning curve* $\bar{L}(t)$ *is non-increasing in* $t$.

*Proof.* Let $M := \lfloor \log_2 T \rfloor$, with $\tau_0 = 0$, $\tau_i = 2^{i-1}$ for $1 \leq i \leq M$, and $\tau_{M+1} = T$. Fix any $\boldsymbol{\mu} \in \mathsf{P}$ and set $a_t := \bar{L}(t)$. Since $\bar{L}(t)$ is non-increasing, the optimal strategy with $S = T$ is to update every time, so

$$\inf_{\Phi} \mathsf{Risk}_{T,T}(\Phi, \boldsymbol{\mu}) = \sum_{t=0}^{T-1} a_t \quad \text{and} \quad \mathsf{Risk}_{M,T}(\Phi_{\mathsf{dyad}}, \boldsymbol{\mu}) = \sum_{i=0}^{M} (\tau_{i+1} - \tau_i)\, a_{\tau_i},$$

where $\Phi_{\mathsf{dyad}}$ denotes the update rule at the $\tau_i$'s. For $i \geq 1$, monotonicity gives

$$2 \sum_{t=\tau_{i-1}}^{\tau_i - 1} a_t \geq 2(\tau_i - \tau_{i-1})\, a_{\tau_i} = 2^{i-1} a_{\tau_i} = (\tau_{i+1} - \tau_i) a_{\tau_i},$$

and for $i = 0$ we have $a_0 \leq 2\sum_{t=0}^{0} a_t$. Summing over $i = 0, \ldots, M$ yields $\mathsf{Risk}_{M,T}(\Phi_{\mathsf{dyad}}, \boldsymbol{\mu}) \leq 2\sum_{t=0}^{T-1} a_t = 2\inf_{\Phi} \mathsf{Risk}_{T,T}(\Phi, \boldsymbol{\mu})$. Taking the supremum over $\boldsymbol{\mu} \in \mathsf{P}$ and applying the minimax inequality completes the proof. $\qquad\square$

**Remark 1.** *The update rule in Theorem 1 resembles the classical "doubling trick" from the online learning literature, which was used to obtain time-independent regret guarantees. However, the level of generality presented here is, to our knowledge, novel. In particular, Theorem 1 is conceptually significant: the update rule requires no knowledge of the exact curve, yet with only exponentially fewer updates, it achieves risk that is competitively optimal w.r.t. the full-update benchmark.*

While Theorem 1 establishes a universal updating rule with only $\log T$ updates, it leaves open the more challenging regime where the update budget is substantially smaller. For illustration, suppose $\bar{L}(t) = O(1/t)$ (e.g., finite-VC $\mathcal{H}$ with an ERM oracle) and only a single update is allowed ($S = 1$). If the update of $\Phi$ is scheduled at time $J$, then $\mathsf{Risk}_{1,T}(\Phi, \boldsymbol{\mu}) \leq O\left(J + \frac{T-J}{J}\right)$, whose minimizer is $J = \sqrt{T}$, yielding cumulative risk of order $\sqrt{T}$.

Generalizing this idea to $S$ updates leads to our next main result:

**Theorem 2.** *Consider any realizable* i.i.d. *learning environment with* $\bar{L}(t) \leq O(1/t)$ *for all* $\boldsymbol{\mu} \in \mathsf{P}$. *If the update times are chosen as* $\tau_i = T^{i/(S+1)}$ *for* $i = 1, \ldots, S$, *then*

$$\mathsf{Risk}_{S,T}(\mathsf{P}) \leq O\big((S+1)T^{1/(S+1)}\big), \qquad S \geq 0.$$

*In particular, for* $S = \log T$, *this recovers the update rule from Theorem 1.*

*Proof.* Let $\Phi$ updates at times $\tau_i = T^{i/(S+1)}$ for $i = 1, \ldots, S$, with $\tau_0 = 0$ and $\tau_{S+1} = T$. Then

$$\mathsf{Risk}_{S,T}(\mathsf{P}) \ \leq \ \sup_{\boldsymbol{\mu} \in \mathsf{P}} \sum_{i=1}^{S+1} (\tau_i - \tau_{i-1}) \, \bar{L}(\tau_{i-1}) \leq \sum_{i=1}^{S+1} \tau_i/\tau_{i-1} = O((S+1)T^{1/(S+1)}). \qquad \square$$

**Remark 2.** *The update rule in Theorem 2 is essentially optimal (up to a factor of $S$) whenever $\bar{L}(t) \geq \Omega(1/t)$. In fact, the optimal risk is lower bounded by $\Omega(T^{1/(S+1)})$. The first update must occur no later than $T^{1/(S+1)}$—otherwise the cumulative risk already exceeds this rate. Similarly, the second update must occur no later than $\tau_1 \cdot T^{1/(S+1)}$, and so on, implying the last update must occur no later than $O(T^{S/(S+1)})$. This in turn forces a risk of $\Omega(T^{1/(S+1)})$ in the final interval.*

Quite surprisingly, when the learning curve follows a power law $\bar{L}(t) \leq 1/t^\alpha$ with $\alpha < 1$, the structure of the optimal scheduling becomes fundamentally different, as shown in the theorem below.

**Theorem 3.** *Consider any realizable* i.i.d. *learning environment with $\bar{L}(t) \leq O(1/t^\alpha)$ for all $\boldsymbol{\mu} \in \mathsf{P}$ and some $\alpha < 1$. Then, for any update budget $S \geq 0$, $\mathsf{Risk}_{S,T}(\mathsf{P}) \ \leq \ O\!\left((S+1)T^{\frac{1-\alpha}{1-\alpha^{S+1}}}\right)$. In particular, with $S = \frac{\log \log T}{\log(1/\alpha)}$ updates we obtain $\mathsf{Risk}_{S,T}(\mathsf{P}) \leq O((S+1)T^{1-\alpha})$.*

*Proof.* Let $\Phi$ update at $\tau_i := T^{u_i}$ with $u_i := \frac{1-\alpha^i}{1-\alpha^{S+1}}$ for $i = 1, \ldots, S+1$, and set $\tau_0 = 0$. Since $\bar{L}(t) \leq O(t^{-\alpha})$, we have for all $\boldsymbol{\mu} \in \mathsf{P}$:

$$\mathsf{Risk}_{S,T}(\Phi, \boldsymbol{\mu}) \ = \ \sum_{i=1}^{S+1} (\tau_i - \tau_{i-1}) \, \bar{L}(\tau_{i-1}) \ \leq \ O\!\left(\sum_{i=1}^{S+1} \frac{\tau_i}{\tau_{i-1}^\alpha}\right).$$

Note that $\tau_i/\tau_{i-1}^\alpha = T^{u_i - \alpha u_{i-1}} = T^{\frac{(1-\alpha)-(\alpha^i - \alpha\alpha^{i-1})}{1-\alpha^{S+1}}} = T^{\frac{1-\alpha}{1-\alpha^{S+1}}}$, we have:

$$\mathsf{Risk}_{S,T}(\Phi, \boldsymbol{\mu}) \ \leq \ \tilde{O}\!\left((S+1)\, T^{\frac{1-\alpha}{1-\alpha^{S+1}}}\right).$$

Taking $S = \frac{\log \log T}{\log(1/\alpha)}$, we have $1 - \alpha^{S+1} \geq 1 - 1/\log T$. Therefore, $T^{\frac{1}{1-\alpha^{S+1}}} \leq T^{\frac{\log T}{\log T - 1}} \leq 2 \cdot T$ for $T \geq 4$. This implies risk $O((S+1)T^{\frac{1-\alpha}{1-\alpha^{S+1}}}) \leq O((S+1)2^{1-\alpha} \cdot T^{1-\alpha}) = O((S+1)T^{1-\alpha})$. $\square$

**Remark 3.** *In fact, for $S = \frac{\log \log T}{\log(1/\alpha)}$, we can even* eliminate *the prefactor $(S+1)$ by using the following two–phase schedule with $\tau_i = 2^{-i} T^{\frac{1-\alpha^i}{1-\alpha^{\lfloor S/2 \rfloor + 1}}}$ for $i = 1, \ldots, \lfloor S/2 \rfloor$ and $\tau_i = \frac{T}{2^{S+1-i}}$ otherwise. This results in $\mathsf{Risk}_{S,T}(\mathsf{P}) \leq O(T^{1-\alpha})$ with $S = O(\log \log T)$.*

For learning curves with $\bar{L}(t) = \Theta(1/t^\alpha)$, the optimal *full-update* risk is of order $\Omega(T^{1-\alpha})$. Theorem 3 shows that with only $S = O(\log \log T)$ updates, the proposed schedule *matches* this full-update risk—with doubly-exponentially fewer updates!

Moreover, both Theorem 2 and Theorem 3 remain near optimal (up to polylogarithmic factors in the risk) even when the learning curve fluctuates within a polylogarithmic factor, i.e., when $\bar{L}(t) = \tilde{\Theta}(1/t)$ and $\bar{L}(t) = \tilde{\Theta}(1/t^\alpha)$, respectively. In fact, for $L(t) = \tilde{\Theta}(1/t^\alpha)$ with $\alpha < 1$, Theorem 3 with $S = O(\log \log T)$ remains *competitively* (near) optimal w.r.t. optimal full-update risk. This is also confirmed by experimental results as shown in Figure 2 and fully discussed in Appendix B.

Unfortunately, the doubly-exponential improvement in the update budget does not extend to the case where the fluctuations are of polynomial size, as shown below. The proof is deferred to Appendix D.

**Theorem 4.** *There exists a non-increasing convex function $\bar{L}(t)$ satisfying $\tilde{\Omega}(t^{-\alpha_1}) \leq \bar{L}(t) \leq \tilde{O}(t^{-\alpha_2})$ for some constants $0 < \alpha_1 < \alpha_2 < 1$, such that for any realizable* i.i.d. *learning environment with learning curve $\bar{L}(t)$ we have*

$$\mathsf{Risk}_{S,T}(\mathsf{P}) \geq T^{\Omega(1/S)} \cdot \mathsf{Risk}_{T,T}(\mathsf{P}).$$

*In particular, for $S \leq o(\frac{\log T}{\log \log T})$, the competitive ratio must be super-polylogarithmic.*

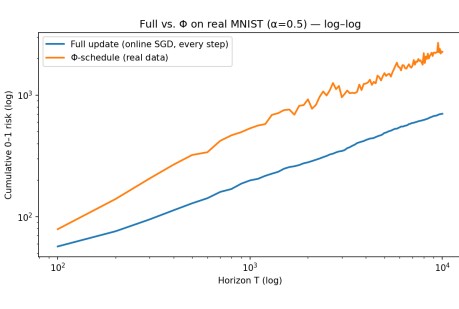 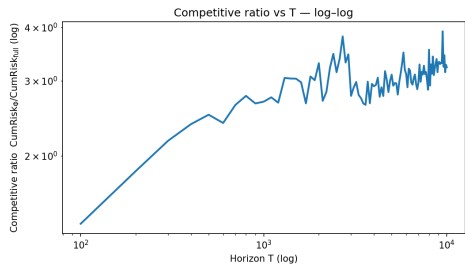

(a) Cumulative risk vs. $T$ (log–log ).    (b) Competitive ration vs. $T$ (log–log).

Figure 2: Comparison of full updates and the $\Phi$-schedule (Theorem 3 with $\alpha = 0.5$) on MNIST, plotted on a log–log scale. With only $S = \Theta(\log \log T)$ updates, the risk remains within a constant factor ($\sim 3 - 4$) of the full-update baseline. See Appendix B for details of the experimental setup.

## 3.2 Risk-Driven Update Schedule

We have established several optimal update schedules that rely only on the shape of the learning curve. In practice, however, the learning curve is rarely known in advance, and the learning environment may deviate substantially from the i.i.d. assumption. In this section, we leverage our theoretical schedules as *primitives*, which can be composed and adapted to design update rules for more realistic settings, thereby bridging the gap between idealized theory and practical applicability.

We begin with the case where the (realized) learning curve $L(t)$ follows a power law with high probability, but with an *unknown* exponent $\alpha$ (e.g., hypothesis classes with finite fat-shattering dimension under ERM retraining). Specifically, we assume that $L(t) = \tilde{\Theta}\big(\frac{1}{t^\alpha}\big)$ holds simultaneously for all $t$ with probability at least $1 - \frac{1}{T}$, for some (unknown) $\alpha > 0$.

Without knowledge of the exponent $\alpha$, the update schedule proposed in Theorem 3 no longer applies. To address this, we introduce Algorithm 1, which monitors the *empirical loss* $\hat{L}(t)$ (defined in Algorithm 1) to guide updates. The algorithm begins with an initial estimate $\hat{\alpha}$ and keeps it fixed as long as the empirical learning curve $\hat{L}(t)$ behaves typically, as guaranteed by Lemma 2 below. When $\hat{L}(t)$ deviates from this typical behavior, $\hat{\alpha}$ is updated according to line 8 of Algorithm 1.

---

**Algorithm 1** Adaptive schedule for unknown $\alpha < 1$

1: **Input:** Time horizon $T$;
2: **Output:** Update rule $\Phi = (\Phi[1], \dots, \Phi[T])$;
3: **Initialize:** Let $\hat{\alpha} = 1 - \frac{\log \log T}{\log T}$, $\hat{S} = \frac{\log \log T}{\log \frac{1}{\hat{\alpha}}}$, $\hat{\beta} = \frac{1-\hat{\alpha}}{1-\hat{\alpha}^{\hat{S}}}$, $i = 0$, $\tau_i = 1$ $\tau_{i+1} = T^{\hat{\beta}}$, and
    randomly pick $\hat{h}_0 \in \mathcal{H}$;
4: **for** $t = 1, \dots, T$ **do**
5:     Compute $\hat{L}(t) = \frac{\sum_{j=\tau_i+1}^{t} \ell(\hat{h}_{\tau_i}(X_j), Y_j)}{t - \tau_i}$. Let $\Phi[t] = 0$;
6:     **if** $t = \tau_{i+1}$ **then**
7:         Compute $\alpha_{max} = \max\{\alpha : \frac{2\hat{L}(t)}{3} \le \frac{C_1 \log \tau_i}{\tau_i^\alpha}\}$;
8:         **if** $\frac{(t-\tau_i)\hat{L}(t)}{14} \ge 2 \log T$ **then**
9:             Let $\hat{\alpha} = \alpha_{max}$, $\hat{S} = \frac{\ln \ln T}{\ln \frac{1}{\hat{\alpha}}}$, and $\hat{\beta} = \frac{1-\hat{\alpha}}{1-\hat{\alpha}^{\hat{S}}}$;
10:        **end if**
11:        Let $\Phi[t] = 1$, $i = i + 1$, $\hat{h}_{\tau_i} = \mathcal{O}(\{X_i, Y_i\}_{i=1}^{\tau_i})$ and $\tau_{i+1} = \max\{2^{-i}\tau_i^{\hat{\alpha}}T^{\hat{\beta}}, 2\tau_i\}$;
12:    **end if**;
13: **end for**

---

Our next main result shows that Algorithm 1 is competitively optimal, *agnostic* to $\alpha$. Note that, while the fluctuations of $L(t)$ are logarithmic in the theorem and in Algorithm 1, the algorithm and its proof can be easily extended to handle polylogarithmic fluctuations.

**Theorem 5.** *Assume there exist constants $C_1 \geq C_2 > 0$, and an unknown exponent $\alpha \in [\Omega(\frac{\log \log \log T}{\log \log T}), 1 - O(\frac{\log \log T}{\log T})]$ such that the realized learning curve $L(t)$ of the oracle satisfies $C_2\, t^{-\alpha} \leq \bar{L}(t) \leq C_1 \log t \cdot t^{-\alpha}$. Then Algorithm 1 makes at most $S = O\left(\frac{\log \log T}{\log(1/\alpha)}\right)$ updates and attains cumulative risk $\mathsf{Risk}_{S,T}(\mathsf{P}) \leq \tilde{O}(T^{1-\alpha})$.*

*Sketch of Proof.* At a high level, Algorithm 1 estimates the true exponent $\alpha$ from the empirical loss $\hat{L}(t)$, and then uses the current estimate $\hat{\alpha}$ to set the update schedule similar to that in Remark 3. Let $\tau_{i_1}, \ldots, \tau_{i_J}$ be the times when the **if** condition in line 8 of Algorithm 1 is triggered, i.e., when $\hat{\alpha}$ is updated. We show that, with probability at least $1 - \frac{1}{T}$, for all $t \geq \tau_{i_j}$ and all $j = 1, \ldots, J$, $\hat{\alpha} \geq \alpha \geq \hat{\alpha} - \frac{\log \log \tau_{i_j-1} + \log(3C_1/C_2)}{\log \tau_{i_j-1}}$. Thus $\hat{\alpha}$ never underestimates $\alpha$, and the gap shrinks as $\tau_{i_j}$ grows. We then show that the *upper estimate* yields a risk bound $\tilde{O}(T^{1-\alpha})$ by our schedule design, while the *lower bracket* on $\alpha$ ensures that the number of updates satisfies $S = O\left(\frac{\log \log T}{\log(1/\alpha)}\right)$. See Appendix E for the complete proof. $\square$

### 3.2.1 PIECEWISE-STATIONARY ENVIRONMENT

We now turn to piecewise-stationary environments, where the data distribution may *change* over time. Formally, there are $K$ *unknown* change points $1 < \kappa_1 < \cdots < \kappa_K \leq T$, with $\kappa_0 = 1$ and $\kappa_{K+1} = T + 1$. Within each segment $t \in [\kappa_{i-1}, \kappa_i)$ the samples $\{(X_t, Y_t)\}$ are i.i.d. from some (unknown) distribution $\mu^{(i)}$, while the distributions may differ across segments.

For simplicity, we focus on binary classification with a hypothesis class $\mathcal{H} \subseteq \{0,1\}^{\mathcal{X}}$ of finite VC dimension and an ERM oracle. We further assume *segment-wise realizability*: for all $i \leq K$, there exists $h_i^\star \in \mathcal{H}$ such that $h_i^\star(X_t) = Y_t$ almost surely for all $t \in [\kappa_{i-1}, \kappa_i)$.

We begin with the simpler benchmark setting in which the change points are *known* to the scheduler.

**Proposition 1.** *Assume the change points $\kappa_1, \ldots, \kappa_K$ are known. Allocate $m := \lfloor S/(K+1) \rfloor$ updates per segment and apply the schedule from Theorem 2 within each segment, we have*

$$\mathsf{Risk}_{S,T}(\mathsf{P}) \leq \tilde{O}\left((S + K)\left(\frac{T}{K+1}\right)^{\frac{1}{\lfloor S/(K+1) \rfloor + 1}}\right).$$

*Sketch of Proof.* Within segment $k$ of length $n_k$, Theorem 2 (using data *only* in that segment) gives a per-segment risk of order $(m+1)\, n_k^{1/(m+1)} \log n_k$. Summing over segments,

$$\sum_{k=1}^{K+1} (m+1)\, n_k^{1/(m+1)} \log n_k \leq (m+1) \log T \sum_{k=1}^{K+1} n_k^{1/(m+1)}.$$

Since $x \mapsto x^{1/(m+1)}$ is concave, Jensen's inequality yields

$$\sum_{k=1}^{K+1} n_k^{1/(m+1)} \leq (K+1)\left(\frac{1}{K+1} \sum_{k=1}^{K+1} n_k\right)^{1/(m+1)} = (K+1)\left(\frac{T}{K+1}\right)^{1/(m+1)}.$$

Combining and using $(K+1)(m+1) \leq S+K+1 = O(S+K)$ gives the stated bound. $\square$

The more challenging regime is when the change points are *unknown*: the scheduler must choose both the update times and which samples to pass to the retraining oracle. Algorithm 2 addresses this issue by monitoring the empirical losses to detect shifts. In Theorem 6 we show that, with high probability, it achieves the same risk (up to polylog factors) as the known–change-point benchmark. The proof is deferred to Appendix F.

**Theorem 6.** *Let $\mathsf{P}$ be piecewise-stationary i.i.d. processes with $K$ change points (unknown to the learner). Assume binary classification with a finite-VC class $\mathcal{H} \subseteq \{0,1\}^{\mathcal{X}}$ and an ERM oracle, i.e.,*

---

**Algorithm 2** Adaptive schedule for piecewise-stationary environment

---

1: **Input:** Time horizon $T$

2: **Initialize:** Let $j = 0$, $i = 1$, $\tau_0 = 0$, $\tau_1 = \left(\frac{T}{K+1}\right)^{\frac{1}{\left\lfloor\frac{S}{K+1}\right\rfloor+1}}$, $d = 0$, and randomly pick $\hat{h}_0 \in \mathcal{H}$;

3: /* $j$ is the number of updates since the last distribution change detected by the **if** condition 6. $\tau_{i-1}$ and $\tau_i$ are the last and next time for model update, respectively. $d$ is used to guarantee that $\tau_i$ cannot be changed once given. */

4: **for** $t = 1, \ldots, T$ **do**

5:     Compute $\widehat{CL}(t) = \sum_{i=\tau_{i-1}+1}^{t} \ell(\hat{h}_{\tau_{i-1}}(X_t), Y_t)$ and $\epsilon(t) = \widehat{CL}(t) - \frac{C_0 \log T(t-\tau_{i-1})}{\left(\frac{T}{K+1}\right)^{\left\lfloor\frac{S}{K+1}\right\rfloor+1}}$;

6:     **if** $\frac{\epsilon^2(t)}{\frac{2C_0 \log T(t-\tau_{i-1})}{\left(\frac{T}{K+1}\right)^{\left\lfloor\frac{S}{K+1}\right\rfloor+1}} + \frac{2\epsilon(t)}{3}} \geq 2\ln T$ and $d = 0$ **then**

7:         Let $\hat{\kappa} = t$, $\tau_i = t + \left(\frac{T}{K+1}\right)^{\frac{1}{\left\lfloor\frac{S}{K+1}\right\rfloor+1}}$, $d = 1$, and $j = 0$;

8:     **else if** $t = \tau_i$, $i < S$, and $j < \left\lfloor\frac{S}{K+1}\right\rfloor$ **then**

9:         Update model $\hat{h}_t = \mathcal{O}(\{X_i, Y_i\}_{i=\hat{\kappa}}^{t-1})$ and let $j = j+1$, $i = i+1$, $\tau_i = t + \left(\frac{T}{K+1}\right)^{\frac{j+1}{\left\lfloor\frac{S}{K+1}\right\rfloor+1}}$, and $d = 0$;

10:   **end if**

11: **end for**

---

the realized learning curve $L(t)$ satisfies $\bar{L}(t) \leq \frac{C_0 \log(t-\tau_{s(t)})}{t-\tau_{i-1}}$ (see (5)), where $t, \tau_{s(t)} \in [\kappa_{i-1}, \kappa_i)$ for some $i$. Then, with probability at least $1 - \frac{1}{T}$, Algorithm 2 using at most $S$ updates achieves

$$\mathsf{Risk}_{S,T}(\mathsf{P}) \leq \tilde{O}\left((S+K)\left(\frac{T}{K+1}\right)^{\frac{1}{\lfloor S/(K+1)\rfloor+1}}\right).$$

*Sketch of Proof.* At a high level, Algorithm 2 tracks the cumulative empirical loss $\widehat{CL}(t)$ since the last update. The threshold in line 6 is chosen so that, if *no* distributional change has occurred since the last update, then with high probability the condition is not triggered. Thus, when it *is* triggered, a change has (w.h.p.) occurred and we refresh the monitor for $\hat{\kappa}$. The values of $\hat{\kappa}$ effectively partition the horizon into at most $K+1$ segments, with the update schedule within each segment the same as in the i.i.d. case (using data only from the last change detected). It is still possible that a distribution change occurs but is not detected before the next scheduled update; this does not affect the final bound because the threshold is applied directly to the empirical loss and hence *self-bounds* the loss accrued in that epoch (an "optimistic" case). See Appendix F for the complete proof. $\square$

### 3.3 No Universal Algorithm Exists

In the previous section we proposed several *universal* update schedules that, without exact knowledge of the learning environment, still attain competitive optimality (matching the known-environment benchmark up to polylogarithmic factors). This raises a natural question: how far can we relax prior knowledge about the environment while retaining such optimality?

Our final main result answers this with a *no–free–lunch*–type negative theorem: in the absence of *any* prior information about the learning environment, no risk-driven (empirical-loss–based) universal update rule can be competitively optimal. The proof is deferred to Appendix G.

**Theorem 7.** *There exists a distribution class* $\mathsf{P}$ *and two hypothesis classes* $\mathcal{H}_1$ *and* $\mathcal{H}_2$ *such that any update rule* $\Phi$ *with* $S = 1$ *that lacks knowledge of the hypothesis class (while being allowed knowledge of* $\mathsf{P}$*) incurs* $T^{\Omega(1)}$ *times the optimal expected risk in at least one of the environments* $(\mathsf{P}, \mathcal{H}_1, \mathcal{O}^{\mathcal{H}_1})$ *or* $(\mathsf{P}, \mathcal{H}_2, \mathcal{O}^{\mathcal{H}_2})$*, where* $\mathcal{O}^{\mathcal{H}_1}$ *and* $\mathcal{O}^{\mathcal{H}_2}$ *denote ERM oracles for* $\mathcal{H}_1$ *and* $\mathcal{H}_2$*.*

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

# A ADDITIONAL DISCUSSION

In this paper, we introduced a general framework for analyzing the *optimal scheduling* of retraining in online models. Our results provide the first precise characterizations of when and how often retraining is needed: we designed schedule rules tailored to the shape of the learning curve, proved they achieve fundamental performance limits, and extended them to more realistic regimes by leveraging empirical losses to guide updates. Together, these results move beyond heuristic criteria and establish a principled theory of retraining frequency.

For clarity of exposition, we adopted several simplifying assumptions. For instance, we treated the retraining budget as a single oracle call, independent of the size of the training set. In practice, the computational cost of retraining often depends heavily on the volume of data processed. A natural future direction is to investigate optimal update schedules under joint constraints of update frequency and data cost. Another promising extension is to adapt our framework to broader learning environments, such as smooth adversarial processes (Haghtalab et al., 2020), where nonstationarity is more structured yet still adversarial in spirit. Finally, our impossibility result highlights a fundamental tension: without minimal prior knowledge of the learning environment, no universal update rule can guarantee optimal performance. This opens an exciting frontier for future work—identifying the weakest possible assumptions under which efficient, risk-driven update rules are still feasible.

# B EMPIRICAL VALIDATION

**Data and models.** We primarily report results on **MNIST**, streamed in a single random order (uniform shuffle of the train split, fixed across horizons). The predictor is a two-layer MLP with 256 hidden units and ReLU activations, followed by a linear 10-way classification head. Inputs are normalized using the standard mean/variance.

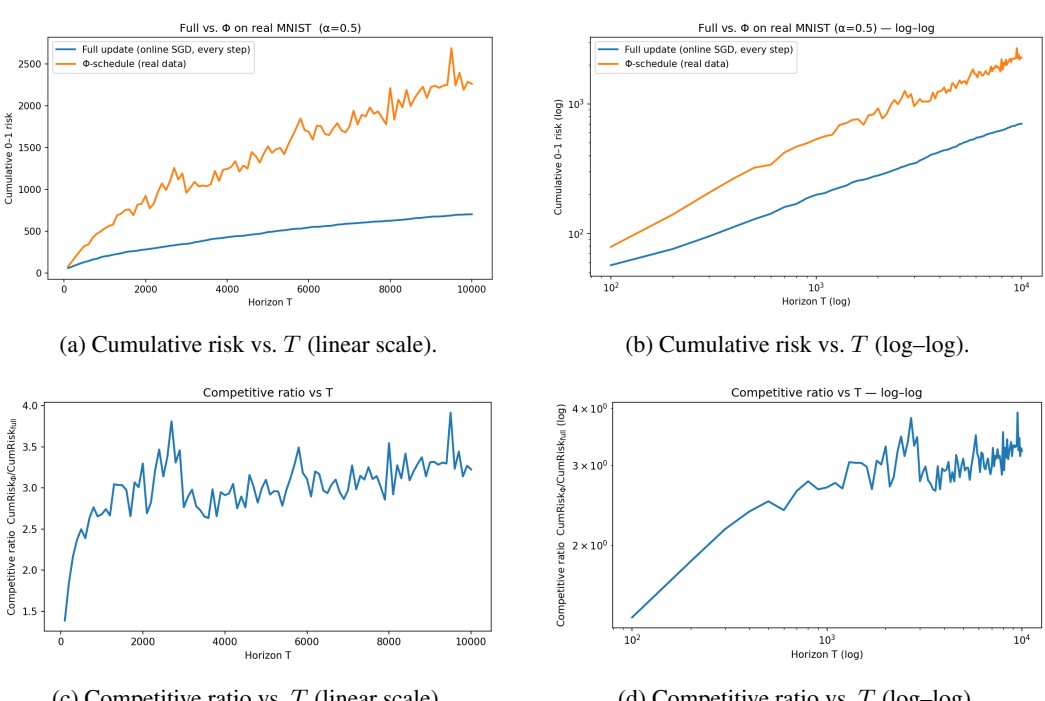

(a) Cumulative risk vs. $T$ (linear scale).

(b) Cumulative risk vs. $T$ (log–log).

(c) Competitive ratio vs. $T$ (linear scale).

(d) Competitive ratio vs. $T$ (log–log).

Figure 3: **Full update vs. $\Phi$-schedule on MNIST** ($\alpha = 0.5$). (a–b) Cumulative 0–1 risk under each policy. The $\Phi$-schedule, despite performing only $S(T) = \Theta(\log \log T)$ updates, remains within a constant-factor of the full-update baseline. (c–d) Competitive ratio. The ratio grows slowly with $T$ (reaching $\sim 3$–$4\times$ at $T = 10^4$), consistent with the predicted power-law learning-curve model.

**Evaluation protocol.** We follow the standard online prediction paradigm. At each round $t$, the learner receives an input $X_t$, produces a prediction $\hat{Y}_t$ *before update*, and incurs instantaneous 0–1 loss $\mathbf{1}[\hat{Y}_t \neq Y_t]$. The example $(X_t, Y_t)$ is then added to the training buffer, after which the learner may update its model depending on the chosen schedule. Thus every example is first tested and only subsequently used for training.

We compare two update schedules:

- **Full update.** At every round $t$, after prediction, the model performs a single stochastic gradient step (AdamW, learning rate $5 \times 10^{-4}$, weight decay $10^{-2}$) on a mini-batch of size 256 sampled from the prefix buffer $\{(X_i, Y_i)\}_{i \leq t}$. This baseline consumes $O(T)$ updates.

- **$\Phi$-schedule.** The model is updated only at the following checkpoints (as in Theorem 3)

$$\tau_i = \lfloor T^{u_i} \rfloor, \qquad u_i = \frac{1 - \alpha^i}{1 - \alpha^{S(T)+1}}, \quad i = 1, \dots, S(T) + 1,$$

where $S(T) = \lfloor \frac{\log \log T}{\log(1/\alpha)} \rfloor$. Unless stated otherwise we use $\alpha = 0.5$ (can be tuned), so

$$S(T) = \Theta(\log \log T).$$

At each checkpoint $t \in \{\tau_i\}$ the model is retrained for two epochs over the entire prefix buffer $\{(X_i, Y_i)\}_{i \leq t}$ (batch size 256). Between checkpoints *no* updates are performed.

We run the evaluation protocol independently for horizons $T \in [10^2, 10^4]$ in increments of 100, using the same shuffled data stream prefix for comparability. All experiments were run on an HPC system with two NVIDIA H200 GPUs.

**Metrics.** Our primary metric is the *cumulative risk*

$$\text{CumRisk}(T) = \sum_{t=1}^{T} \mathbf{1}[\hat{Y}_t \neq Y_t],$$

measured on pre-updated model predictions. We also report the *competitive ratio*

$$\text{CumRisk}_\Phi(T) / \text{CumRisk}_{\text{full}}(T).$$

**Findings.** From Figure 3 we see that with only $\Theta(\log \log T)$ retrainings, $\Phi$ attains cumulative risk within a bounded factor (around $3 \sim 4 \times$ for $T = 10^4$) of full update, demonstrating substantial savings in update cost.

**Fairness of comparison.** Both rules are evaluated under the same "predict before update" protocol: every example contributes to the risk before it can influence the model. The distinction is in *latency*: the full-update learner incorporates each new sample immediately (after its prediction), while the $\Phi$-schedule may defer incorporation until the next checkpoint, at which point the entire prefix is used. This exactly captures the intended frequency–accuracy trade-off.

## C   LOWER BOUND ON POWER LAW LEARNING CURVE

In the next example, we construct a hypothesis class $\mathcal{H}$ based on the Haar measure whose learning curve exhibits a power-law *lower bound* that, up to a $\log T$ factor, matches the *upper bound* implied by (equation 6). We do this for every rational positive $\alpha = \frac{p}{q} \leq 1$, $p, q \in \mathbb{Z}_+$. For any irrational $\alpha$, one can find a rational $\alpha'$ such that $|\alpha - \alpha'| \leq \frac{1}{\log T}$ and use the hypothesis class constructed for $\alpha'$. Since $\frac{1}{2} \geq \frac{t^{\alpha'}}{t^\alpha} \leq 2$, the bound implied by (equation 6) for $\alpha'$ works for $\alpha$.

**Example 1.** *Let* $I_{l,j} = (\frac{j-1}{2^{ql}}, \frac{j}{2^{ql}}]$, $l \in \mathbb{Z}_+$, $j \in \{1, \dots, 2^{ql}\}$ *be* $2^{ql}$ *intervals that uniformly split* $(0, 1]$. *In addition, let* $g_{l,j}(x) = \sum_{i=2^{q-1}}^{2^q - 1} \mathbb{1}(x \in I_{l+1, 2^q j - i}) - \sum_{i=0}^{2^{q-1}-1} \mathbb{1}(x \in I_{l+1, 2^q j - i})$ *for* $l \in \mathbb{Z}_+$ *and* $j \in \{1, \dots, 2^{ql}\}$. *Define the hypothesis class*

$$\mathcal{H} = \left\{ h(x) = \sum_{l=1}^{\infty} \sum_{j=1}^{2^{ql}} c_l \sigma_{l,j} g_{l,j}(x) : \sigma_{l,j} \in \{-1, 1\} \right\} \tag{8}$$

where $c_l = \frac{1}{2^{\frac{pl}{2}}}$, $l \in \mathbb{Z}_+$. Let $X_1, \ldots, X_T \sim U(0,1)$ be i.i.d. *random variables uniformly distributed over* $(0,1)$. *Let $\ell$ be the square loss. For the ground truth function $h \in \mathcal{H}$, let*

$$\mathcal{O}(\{X_i, h(X_i)\}_{i=1}^t) = \arg \min_{\hat{h} \in \mathcal{H}: \hat{h}(X_i) = h(X_i), i \in \{1,\ldots,T\}} \sup_{h \in \mathcal{H}} \mathbb{E}_{X \sim U(0,1)}[(h(X) - \hat{h}(X))^2]$$

*return the ERM that minimizes the worst-case risk. If there are multiple such ERMs, $\mathcal{O}(\{X_i, h(X_i)\}_{i=1}^t)$ returns the minimum $(\sigma_{l,j} : l \in \mathbb{Z}_+, j \in \{1,\ldots,2^{ql}\})$ in lexicographic order.*

**Lemma 1.** *For the hypothesis class $\mathcal{H}$, data distribution $X_i \sim U(0,1)$, and oracle $\mathcal{O}$ given in Example 1, there exists $\alpha$ and a ground truth function $f \in \mathcal{H}$ such that $\bar{L}(t) \leq O(t^{-\alpha})$ with probability at least $1 - t^{-2}$.*

We start with the following result.

**Proposition 2.** *Let $I_{l,j_{l,i}}$, $l \in \mathbb{Z}_+$, $i \in \{1,\ldots,t\}$ be the collection of intervals that contain $X_i$. There is a unique set of $\hat{\sigma}_{l-1,\lceil \frac{j_{l,i}}{2^q} \rceil} \in \{-1,1\}$ such that $\sum_{l=1}^\infty c_l \hat{\sigma}_{l,\lceil \frac{j_{l+1,i}}{2^q} \rceil} g_{l,\lceil \frac{j_{l+1,i}}{2^q} \rceil}(x) = h(X_i)$. Moreover, we have*

$$\mathcal{O}(\{X_i, h(X_i)\}_{i=1}^t)$$
$$= \sum_{i=1}^t \sum_{l=1}^\infty \left( \mathbb{1}((l,j) \in W_t) c_l \hat{\sigma}_{l,\lceil \frac{j_{l+1,i}}{2^q} \rceil} g_{l,\lceil \frac{j_{l+1,i}}{2^q} \rceil} - \mathbb{1}((l,j) \notin W_t) c_l g_{l,\lceil \frac{j_{l+1,i}}{2^q} \rceil} \right) \quad (9)$$

*and*

$$\mathbb{E}_{X \sim U(0,1)}[(h(X) - \mathcal{O}(\{X_i, h(X_i)\}_{i=1}^t))^2]$$
$$= \sum_{l=1}^\infty \sum_{j=1}^{2^{ql}} \mathbb{1}(I_{l,j} \cap \{X_i\}_{i=1}^t = \emptyset \text{ and } I_{l-1,\lceil \frac{j}{2^q} \rceil} \cap \{X_i\}_{i=1}^t \neq \emptyset) \frac{(\sum_{u=l}^\infty c_l \sigma_{l,j} + 1)^2}{2^{ql}}$$
$$= \sum_{l=1}^\infty \sum_{j=1}^{2^{ql}} \mathbb{1}(I_{l,j} \cap \{X_i\}_{i=1}^t = \emptyset) \frac{c_l^2}{2^{ql}} (\sigma_{l,j} + 1)^2, \quad (10)$$

*where $W_t = \{(l, \lceil \frac{j_{l+1,i}}{2^q} \rceil) : l \in \mathbb{Z}_+, i \in \{1,\ldots,t\}\}$.*

*Proof.* Let

$$\mathcal{O}(\{X_i, h(X_i)\}_{i=1}^t) = \sum_{l=1}^\infty \sum_{j=1}^{2^{ql}} c_l \hat{\sigma}_{l,j} g_{l,j}(x).$$

and

$$h(x) = \sum_{l=1}^\infty \sum_{j=1}^{2^{ql}} c_l \sigma_{l,j} g_{l,j}(x). \quad (11)$$

We first show that $\hat{\sigma}_{l,\lceil \frac{j_{l+1,i}}{2^q} \rceil} = \sigma_{l,\lceil \frac{j_{l+1,i}}{2^q} \rceil}$, $l \in \mathbb{Z}_+$, $i \in \{1,\ldots,t\}$. Note that $g_{l',j'}(X_i) = 0$ for $(l',j') \notin \{(l, \lceil \frac{j_{l+1,i}}{2^q} \rceil) : l \in \mathbb{Z}_+\}$ since $\cup_{i=0}^{2^q-1} I_{l'+1,2^q j'-i}$ does not contain $X_i$. Hence, we have

$$h(X_i) = \sum_{l=1}^\infty \sigma_{l,\lceil \frac{j_{l+1,i}}{2^q} \rceil} g_{l,\lceil \frac{j_{l+1,i}}{2^q} \rceil}(x) c_l.$$

On the other hand, the choice of values of $\sigma_{l,\lceil \frac{j_{l+1,i}}{2^q} \rceil}$, $l \in \mathbb{Z}_+$ is unique since otherwise, we have

$$\sum_{l=1}^\infty (\sigma_{l,\lceil \frac{j_{l+1,i}}{2^q} \rceil} - \sigma_{l,\lceil \frac{j_{l+1,i}}{2^q} \rceil}) g_{l,\lceil \frac{j_{l+1,i}}{2^q} \rceil}(x) c_l = \sum_{l=1}^\infty a_l C \left(\frac{1}{2}\right)^{\frac{pl}{2}} = 0,$$

where $a_l = (\sigma_{l,\lceil \frac{j_{l+1,i}}{2^q} \rceil} - \sigma_{l,\lceil \frac{j_{l+1,i}}{2^q} \rceil}) g_{l,\lceil \frac{j_{l+1,i}}{2^q} \rceil}(x) \in \{-2,0,2\}$. Then $\frac{1}{\sqrt{2}}$ is a root of a polynomial of coefficients in $\{-2,0,2\}$ and thus a polynomial of coefficients in $\{-1,0,1\}$, which is impossible since $\sum_{j=1}^d a_j \left(\frac{1}{2}\right)^j = \frac{\sum_{j=0}^{\lfloor \frac{d-1}{2} \rfloor} a_{2j+1} \frac{1}{2^j}}{\sqrt{2}} + \sum_{j=0}^{\lfloor \frac{d}{2} \rfloor} a_{2j} \frac{1}{2^j}$ and $2^i > \sum_{v=0}^{i-1} 2^v$ for any $i \in \mathbb{Z}_+$.

Then, since $\int_0^1 g_{l,j}(x) g_{l',j'}(x) dx = 0$ for $(l,j) \neq (l',j')$ and $\int_0^1 g_{l,j}(x) g_{l,j}(x) dx = \frac{1}{2^{ql}}$, $l \in \mathbb{Z}_+$, we have

$$\mathbb{E}_{X \sim U(0,1)}[(h(X) - \mathcal{O}(\{X_i, h(X_i)\}_{i=1}^t))^2]$$

$$= \sum_{l=1}^\infty \sum_{j=1}^{2^{ql}} \left( \mathbb{1}((l,j) \in W_t) \frac{c_l^2}{2^{ql}} (\hat{\sigma}_{l,j} - \sigma_{l,j})^2 + \mathbb{1}((l,j) \notin W_t) \frac{c_l^2}{2^{ql}} (\hat{\sigma}_{l,j} - \sigma_{l,j})^2 \right)$$

$$= \sum_{l=1}^\infty \sum_{j=1}^{2^{ql}} \mathbb{1}((l,j) \notin W_t) \frac{c_l^2}{2^{ql}} (\hat{\sigma}_{l,j} - \sigma_{l,j})^2, \tag{12}$$

Note that $\mathcal{O}(\{X_i, h(X_i)\}_{i=1}^t)(X_i) = h(X_i)$, $i \in \{1,\ldots,t\}$ for all choices of $\hat{\sigma}_{l,j} \in \{-1,1\}$, $(l,j) \notin W_t$. Every choice has the same minimum worst-case expected loss since $\sigma_{l,j}$ can flip the sign of $\hat{\sigma}_{l,j}$. Therefore, the oracle chooses $\hat{\sigma}_{l,j} = -1$ for $(l,j) \notin W_t$, and we have (9). Note that $(l,j) \in W_t$ only when $I_{l,j} \cap \{X_1,\ldots,X_t\} \neq \emptyset$. Combining (9) and (12), we have (10). $\square$

**Corollary 1.** *For the hypothesis class $\mathcal{H}$, data distribution $X_i \sim U(0,1)$, and oracle $\mathcal{O}$ given in Example 1, there exists a ground truth function $f \in \mathcal{H}$ such that*

$$\bar{L}(t) \leq O(\frac{1}{t^\alpha})$$

*with probability at least $1 - \frac{1}{t^2}$.*

*Proof.* Choose $f$ as in (11) such that $\sigma_{l,j} = 1$ for $l \in \mathbb{Z}_+$, $j \in \{1,\ldots,2^{ql}\}$. Then according to (10), we have

$$\bar{L}(t) = \sum_{l=1}^\infty \frac{4}{2^{pl}} \left(1 - \frac{1}{2^{ql}}\right)^t$$

$$\leq \sum_{l=1}^{\frac{\ln t - \ln \ln t}{\ln 2^q}} \frac{4}{2^{pl}} \left(1 - \frac{1}{2^{ql}}\right)^t + \sum_{l=\frac{\ln t - \ln \ln t}{\ln 2^q}+1}^\infty \frac{4}{2^{pl}} \left(1 - \frac{1}{2^{ql}}\right)^t \tag{13}$$

$$\leq \frac{2(\frac{\ln t - \ln \ln t}{\ln 2^q})}{t^\alpha} + \frac{4\ln^\alpha t}{t^\alpha} \leq O(\frac{\log t}{t^\alpha}). \tag{14}$$

$\square$

## D  PROOF OF THEOREM 4

*Proof.* Let $\tilde{\alpha}_i = \alpha_1 + \frac{i(\alpha_2 - \alpha_1)}{S+1}$ for $i \in \{0,\ldots,S+1\}$. Let $T_{1,i} = T^{\frac{1-\alpha_2}{1-\tilde{\alpha}_i} - \frac{S+1-i}{\log T}}$ for $i \in \{1,\ldots,S+1\}$ and let $T_{1,0} = 1$. Let $(T_{2,i}, \frac{1}{T_{2,i}^{\tilde{\alpha}_{i+1}}})$, $i \in \{0,\ldots,S\}$, be the intersection of the tangent line of the curve $f_i(x) = \frac{1}{x^{\tilde{\alpha}_i}}$ at point $(T_{1,i}, \frac{1}{T_{1,i}^{\tilde{\alpha}_i}})$ and the curve $f_{i+1}(x) = \frac{1}{x^{\tilde{\alpha}_{i+1}}}$ such that $T_{2,i} > T_{1,i}$, i.e., we have

$$\frac{-\tilde{\alpha}_i}{T_{1,i}^{\tilde{\alpha}_i+1}}(T_{2,i} - T_{1,i}) + \frac{1}{T_{1,i}^{\tilde{\alpha}_i}} = \frac{1}{T_{2,i}^{\tilde{\alpha}_{i+1}}}. \tag{15}$$

From (15) we have that $\frac{1+\tilde{\alpha}_i}{\tilde{\alpha}_i} T_{1,i} > T_{2,i}$. Therefore,

$$T_{1,i+1} = O(T^{\frac{1-\alpha_2}{(S+1)(1-\tilde{\alpha}_i)(1-\tilde{\alpha}_{i+1})}} T_{1,i}) \geq \frac{1+\tilde{\alpha}_i}{\tilde{\alpha}_i} T_{1,i} > T_{2,i}$$

for $i \in \{0, \ldots, S\}$ and $S = o(\log T)$. Let

$$
L(t) = \begin{cases} \frac{-\tilde{\alpha}_i}{T_{1,i}^{\tilde{\alpha}_i+1}}(t - T_{1,i}) + \frac{1}{T_{1,i}^{\tilde{\alpha}_i}} & T_{1,i} \le t \le T_{2,i} \text{ for some } i \in \{0, \ldots, S\}, \\ \frac{1}{t^{\tilde{\alpha}_i+1}} & T_{2,i} \le t \le T_{1,i+1} \text{ for some } i \in \{0, \ldots, S\}. \end{cases} \tag{16}
$$

Then, we have

$$
\inf_{\Phi} \mathrm{Risk}_{T,T}(\Phi, \boldsymbol{\mu}) = \sum_{t=0}^{T-1} L(t) \le \sum_{t=1}^{T_{2,0}} \frac{1}{t^{\tilde{\alpha}_0}} + \sum_{i=0}^{S-1} \sum_{t=T_{2,i}+1}^{T_{2,i+1}} \frac{1}{t^{\tilde{\alpha}_{i+1}}} + \sum_{t=T_{S,2}+1}^{T} \frac{1}{t^{\tilde{\alpha}_{S+1}}}
$$

$$
= \sum_{i=0}^{S+1} O(T_{2,i}^{1-\tilde{\alpha}_i})
$$

$$
= \sum_{i=0}^{S+1} O(T_{1,i}^{1-\tilde{\alpha}_i})
$$

$$
= O(T^{1-\alpha_2}).
$$

Since there are at most $S$ model updates, there exists some $i \in \{0, \ldots, S\}$ such that there is no model update in the time interval $T_{1,i}$ and $T_{1,i+1}$. Therefore,

$$
\mathrm{Risk}_{S,T}(\mathsf{P}) \ge \frac{T_{1,i+1} - T_{1,i}}{T^{\tilde{\alpha}_i}} \ge T^{\frac{(1-\alpha_2)(1-\tilde{\alpha}_i - \tilde{\alpha}_i(1-\tilde{\alpha}_{i+1}))}{(1-\tilde{\alpha}_i)(1-\tilde{\alpha}_{i+1})} + \frac{(S+1)(1-\tilde{\alpha}_i)}{\log T}} = T^{1-\alpha_2 + O(\frac{1}{S}) + O(\frac{S}{\log T})}.
$$

Hence when $S = o(\log T)$, we have $\mathrm{Risk}_{T,T}(\mathsf{P}) = o(\mathrm{Risk}_{S,T}(\mathsf{P}))$. $\qquad\square$

# E  PROOF OF THEOREM 5

We start with the following well known large deviation result.

**Lemma 2.** *For a realizable* i.i.d. *learning environment, let $\hat{L}(t)$ defined in Algorithm 1. For any $C > 0$, we have*

$$
Pr(|\hat{L}(t) - L(t)| > C \cdot L(t)) \le \exp\left(-\frac{C^2(t - \tau_{s(t)})L(t)}{2 + \frac{2C}{3}}\right), \tag{17}
$$

*where $\tau_{s(t)}$ is the time of the most recent model update up to time $t$.*

*Proof.* (of Theorem 5) Let $\tau_{i_1}, \ldots, \tau_{i_J}$ be the time when the **if** condition in line 8 of Algorithm 1 is satisfied after the $\tau_i$ model update and when $\hat{\alpha}$ is upgraded. We first show that with probability at least $1 - \frac{1}{T}$, we have $\hat{\alpha} \ge \alpha \ge \hat{\alpha} - \frac{\log\log\tau_{i_j-1} + \log(\frac{3C_1}{C_2})}{\log\tau_{i_j-1}}$ for $t \ge \tau_{i_j}$, $j \in \{1, \ldots, J\}$. By our assumption we have $\bar{L}(t) \le \frac{C_1 \log \tau_{s(t)}}{\tau_{s(t)}^{\alpha}}$ and $\bar{L}(t) \ge \frac{C_2}{\tau_{s(t)}^{\alpha}}$ for some positive constant $C_1$ and $C_2$. To this end, with high probability we have $\frac{\bar{L}(\tau_{i_j}-1)}{2} \le \hat{L}(\tau_{i_j}) \le \frac{3\bar{L}(\tau_{i_j}-1)}{2}$. Let $L_i^* = \frac{28\log T}{\tau_i - \tau_{i-1}}$, $i \in \{1, \ldots, S\}$ be the minimum value of $\hat{L}(t)$ so that the **if** condition in line 8 in Algorithm 1 is satisfied. Then according to Lemma 2, for $\bar{L}(\tau_{i_j-1}) \le \frac{2L_{i_j}^*}{3}$ we have (setting $C = L_{i_j}^*/(\bar{L}(\tau_{i_j-1})-1)$ in Lemma 2)

$$
\Pr(\hat{L}(\tau_{i_j}) \ge L_{i_j}^*) \le \exp\left(-\frac{(\frac{L_{i_j}^*}{\bar{L}(\tau_{i_j-1})} - 1)^2 (\tau_{i_j} - \tau_{i_j-1}) \frac{L_{i_j}^*}{\frac{L_{i_j}^*}{\bar{L}(\tau_{i_j-1})}}}{2 + \frac{2(\frac{L_{i_j}^*}{\bar{L}(\tau_{i_j-1})}-1)}{3}}\right)
$$

$$
\overset{(a)}{\le} \exp\left(-\frac{(\tau_{i_j} - \tau_{i_j-1})L_{i_j}^*}{14}\right) \le \frac{1}{T^2},
$$

where $(a)$ follows since the function $\frac{x^2}{(1+x)(2+\frac{2}{3x})}$ increases with $x$ for $x \geq \frac{1}{2}$. Similarly, for $\bar{L}(\tau_{i_j-1}) \geq \frac{2L^*_{i_j}}{3}$, we have

$$\Pr(|\hat{L}(\tau_{i_j}) - \bar{L}(\tau_{i_j-1})| \geq \frac{\bar{L}(\tau_{i_j-1})}{2}) \leq \frac{1}{T^2}. \tag{18}$$

Let $M_j$, $j \in \{1, \ldots, J\}$ be the event when $\frac{2\hat{L}(t)}{3} \leq \bar{L}(\tau_{i_j-1}) \leq 2\hat{L}(\tau_{i_j})$ for $j \in \{1, \ldots, J\}$. Then from (18) we have

$$\Pr(\cap_{j=1}^J M_j) \geq 1 - \frac{1}{T}.$$

We then show that $\hat{\alpha} \geq \alpha \geq \hat{\alpha} - \frac{\log\log\tau_{i_j-1}+\log(\frac{3C_1}{C_2})}{\log\tau_{i_j-1}}$ at time $\tau_{i_j}$, $j \in \{1, \ldots, J\}$, when $\cap_{j=1}^J M_j$ occurs. The definition of $\alpha_{max}$ in line 7 in Algorithm 1 and the fact that $\frac{2\hat{L}(\tau_{i_j})}{3} \leq \bar{L}(\tau_{i_j-1}) \leq \frac{C_1\log\tau_{i_j-1}}{\tau_{i_j-1}^\alpha}$ imply that $\alpha_{max} \geq \alpha$. Moreover, $2\hat{L}(\tau_{i_j}) \geq \bar{L}(\tau_{i_j-1}) \geq \frac{C_2}{\tau_{i_j-1}^\alpha}$ and $\frac{2\hat{L}(\tau_{i_j})}{3} \leq \bar{L}(\tau_{i_j-1}) \leq \frac{C_1\log\tau_{i_j-1}}{\tau_{i_j-1}^\alpha}$ imply that

$$\alpha_{max} - \alpha \leq \frac{\log(3C_1\log\tau_{i_j-1}) - \log(2\hat{L}(\tau_{i_j-1}))}{\log\tau_{i_j-1}} - \frac{\log(C_2) - \log(2\hat{L}(\tau_{i_j-1}))}{\log\tau_{i_j-1}}$$

$$= \frac{\log\log\tau_{i_j-1} + \log(\frac{3C_1}{C_2})}{\log\tau_{i_j-1}} \tag{19}$$

for $j \in \{1, \ldots, J\}$. Let $E_j$, $j \in \{1, \ldots, J\}$, be the event where $\alpha \leq \hat{\alpha} \leq \alpha + \frac{\log\log\tau_{i_j-1}+\log(\frac{3C_1}{C_2})}{\log\tau_{i_j-1}}$ at time $t = \tau_{i_j}$. Then we have $\Pr(\cap_{j=1}^J E_j) \geq 1 - \frac{1}{T}$ since $\Pr(\cap_{j=1}^J M_j) \geq 1 - \frac{1}{T}$.

We then show that the total risk achieved by Algorithm 1 is at most $\tilde{O}(T^{1-\alpha})$. Let $\tau_1 < \ldots < \tau_S \leq T$ be the model update time specified in Algorithm 1. We show that $\mathbb{E}[\frac{\tau_{i+1}\log\tau_i}{\tau_i^\alpha} | \cap_{j=1}^J E_j] \leq \tilde{O}(\frac{T^{1-\alpha}}{2^i})$. Note that

$$\mathbb{E}[\frac{\tau_{i+1}\log\tau_i}{\tau_i^\alpha} | \cap_{j=1}^J E_j] \leq \log T \left( \mathbb{E}\left[\frac{\tau_{i+1}}{\tau_i^\alpha} | \cap_{j=1}^J E_j\right] \right)$$

$$\leq \log T \left( \mathbb{E}\left[ \max\{\frac{\tau_i^{\hat{\alpha}} T^{\hat{\beta}}}{2^i \tau_i^\alpha}, \frac{2\tau_i}{\tau_i^\alpha}\} \right] \right)$$

$$\overset{(a)}{\leq} \log T \left( \frac{T^{\hat{\alpha}-\alpha+\hat{\beta}}}{2^i} + 2\tau_i^{1-\alpha} \right)$$

$$= \log T \left( \frac{T^{1-\alpha+\frac{\hat{\alpha}^{\hat{s}}(1-\hat{\alpha})}{1-\hat{\alpha}^{\hat{s}}}}}{2^i} + 2\tau_i^{1-\alpha} \right)$$

$$\leq \log T \left( \frac{T^{1-\alpha+\hat{\alpha}^{\hat{s}}}}{2^i} + 2\tau_i^{1-\alpha} \right),$$

where $(a)$ follows since $\hat{\alpha} \geq \alpha$ and $\tau_i \leq T$. Therefore, we have

$$\text{Risk}_{S,T} = \sum_{i=1}^{S+1} \left( \mathbb{E}[(\tau_i - \tau_{i-1})\hat{L}(\tau_i) | \cap_{j=1}^J E_j]\Pr(\cap_{j=1}^J E_j) + \mathbb{E}[(\tau_i - \tau_{i-1})\hat{L}(\tau_i) | (\cap_{j=1}^J E_j)^c]\Pr((\cap_{j=1}^J E_j)^c) \right)$$

$$\leq \frac{T}{T} + \sum_{i=1}^{S+1} \left( \mathbb{E}[(\tau_i - \tau_{i-1})\hat{L}(\tau_i) | \cap_{j=1}^J E_j]\Pr(\cap_{j=1}^J E_j) \right)$$

$$= 1 + \sum_{i=1}^{S+1} \left( \mathbb{E}[(\tau_i - \tau_{i-1})\bar{L}(\tau_{i-1}) | \cap_{j=1}^J E_j]\Pr(\cap_{j=1}^J E_j) \right)$$

$$\overset{(a)}{\leq} 1 + \sum_{i=1}^{S+1} \log T \left( \frac{T^{1-\alpha+\hat{\alpha}^{\hat{s}}}}{2^{i-1}} + 2\tau_{i-1}^{1-\alpha} \right)$$

$$\leq 1 + 4T^{1-\alpha} \log T + 2T^{1-\alpha} \sum_{i=1}^{S+1} \frac{1}{2^{(i-1)(1-\alpha)}}$$

$$= \tilde{O}(T^{1-\alpha}),$$

where $\tau_{S+1} = T$ and $(a)$ follows from the facts that $\tau_S \leq T$ and $\frac{\tau_i}{\tau_S} = \prod_{j=i+1}^{S} \frac{\tau_j}{\tau_{j-1}} \leq \frac{1}{2^{S-i}}$.

Finally, we show that $S \leq O(\frac{\ln \ln T}{\ln \frac{1}{\alpha}})$ with probability at least $1 - \frac{1}{T} - \frac{1}{T^2}$. More specifically, we show that $\tau_{S+1} \geq T$ following the rule $\tau_{S+1} = \max\{2^{-S} \tau_S^{\hat{\alpha}} T^{\hat{\beta}}, 2\tau_S\}$. Let $j_0 = \min_{j:i_j \geq \frac{2 \log \log T}{\log \frac{1}{\alpha}} + \frac{\log \frac{112}{C_2}}{(1-\alpha)} + 2} j$. We show that with probability at least $\frac{1}{T^2}$, $i_{j_0} = \frac{2 \log \log T}{\log \frac{1}{\alpha}} + \frac{112}{C_2(1-\alpha)} j + 2$. Note that

$$\bar{L}\left(\tau_{\frac{2 \log \log T}{\log \frac{1}{\alpha}} + \frac{112}{C_2(1-\alpha)} j+1}\right) \geq \frac{C_2}{\tau_{\frac{2 \log \log T}{\log \frac{1}{\alpha}} + \frac{112}{C_2(1-\alpha)} j+1}^{\alpha}}$$

$$= \frac{C_2 \tau_{\frac{2 \log \log T}{\log \frac{1}{\alpha}} + \frac{112}{C_2(1-\alpha)} j+1}^{1-\alpha}}{\tau_{\frac{2 \log \log T}{\log \frac{1}{\alpha}} + \frac{112}{C_2(1-\alpha)} j+1}}$$

$$\overset{(a)}{\geq} \frac{C_2 2^{(1-\alpha)(1+\frac{2}{\log(\frac{1}{\alpha})} + \frac{\log \frac{112}{C_2}}{(1-\alpha)}) \log \log T}}{\tau_{\frac{2 \log \log T}{\log \frac{1}{\alpha}} + \frac{112}{C_2(1-\alpha)} j+1}}$$

$$\geq \frac{C_2 2^{(1-\alpha)(1+2 \ln 2 \frac{a}{1-a} + \frac{\log \frac{112}{C_2}}{(1-\alpha)}) \log \log T}}{\tau_{\frac{2 \log \log T}{\log \frac{1}{\alpha}} + \frac{112}{C_2(1-\alpha)} j+1}}$$

$$\geq \frac{112 \log T}{\tau_{\frac{2 \log \log T}{\log \frac{1}{\alpha}} + \frac{112}{C_2(1-\alpha)} j+1}}$$

$$\geq 2L^*_{\frac{2 \log \log T}{\log \frac{1}{\alpha}} + \frac{112}{C_2(1-\alpha)} j+1}$$

where $(a)$ follows since $\tau_1 \geq \log T$ and $\tau_i \geq 2^{i-1} \tau_1$ for $i \in \{1, \dots, S\}$. Then according to (18), we have

$$\Pr\left( \hat{L}\left( \tau_{\frac{2 \log \log T}{\log \frac{1}{\alpha}} + \frac{\log \frac{28}{C_2}}{(1-\alpha)} + 2} \right) \leq L^*_{\frac{2 \log \log T}{\log \frac{1}{\alpha}} + \frac{112}{C_2(1-\alpha)} j+1} \right)$$

$$\leq \Pr\left( \hat{L}\left( \tau_{\frac{2 \log \log T}{\log \frac{1}{\alpha}} + \frac{\log \frac{28}{C_2}}{(1-\alpha)} + 2} \right) \leq \frac{\bar{L}\left( \tau_{\frac{2 \log \log T}{\log \frac{1}{\alpha}} + \frac{\log \frac{28}{C_2}}{(1-\alpha)} + 1} \right)}{2} \right) \leq \frac{1}{T^2}$$

and the if condition happens at $i_{j_0} = \frac{2 \log \log T}{\log \frac{1}{\alpha}} + \frac{\log \frac{28}{C_2}}{(1-\alpha)} + 2$ with probability at least $1 - \frac{1}{T^2}$. Therefore, with probability at least $1 - \frac{1}{T^2} - \frac{1}{T}$, $\cap_{j=1}^{J} E_j$ occurs and we have $i_{j_0} = \frac{2 \log \log T}{\log \frac{1}{\alpha}} + \frac{\log \frac{28}{C_2}}{(1-\alpha)} + 2$. Note that $\tau_i \geq 2^{\frac{\log \log T}{\log \frac{1}{\alpha}}}$ for $i \geq \frac{\log \log T}{\log \frac{1}{\alpha}}$. Combining with (19), we have

$$\alpha_{max} \leq \alpha + \frac{\log \log \tau_{i_{j_0}} + \log(\frac{3C_1}{C_2})}{\log \tau_{j_0}}$$

$$\overset{(a)}{\leq} \alpha + \frac{\log \frac{\log \log T}{\log \frac{1}{\alpha}} + \log(\frac{3C_1}{C_2})}{\frac{\log \log T}{\log \frac{1}{\alpha}}} \tag{20}$$

for $i \geq i_{j_0}$, where $(a)$ holds since the function $\frac{\log x + \log(\frac{3C_1}{C_2})}{x}$ decreases with $x$ when $x \geq e$ and $\log T \geq \frac{e}{\alpha}$. In the following, we show that $\alpha_{max} \leq 3\alpha$ for $t \geq \tau_i$ and $i \geq i_{j_0}$, by considering the cases when $\alpha \geq \frac{1}{2}$ and $\alpha < \frac{1}{2}$ separately. When $\alpha \geq \frac{1}{2}$, let $\alpha = 1 - \frac{C \log \log T}{\log T}$ for $1 \leq C \leq \frac{\log T}{2 \log \log T}$. We have $\log \frac{1}{\alpha} \leq \frac{1-\alpha}{\alpha \ln 2} \leq \frac{2(1-\alpha)}{\ln 2} = \frac{2C \log \log T}{\ln 2 \log T}$. Then according to (20), we have

$$\alpha_{max} \leq \alpha + \frac{\log\left(\frac{\ln 2 \log T}{2C}\right)}{\frac{\ln 2 \log T}{2C}}$$

$$= 1 - \frac{2C \log \log T}{\log T} - \frac{2C \log(\frac{2C}{\ln 2})}{\ln 2} + \frac{2C \log \log T}{\ln 2 \log T}$$

$$\leq 3\alpha.$$

Similarly, when $\alpha < \frac{1}{2}$, let $\alpha = \frac{C \log \log \log T}{\log \log T}$, where $1 \leq C \leq \frac{\log \log T}{2 \log \log \log T}$. We have $\frac{\log \log T}{\log \frac{1}{\alpha}} = \frac{\log \log T}{\log \log \log T - \log C \log \log \log T}$. Then according to (20),

$$\alpha_{max} \leq \alpha + \frac{\log\left(\frac{\log \log T}{\log \log \log T - \log C \log \log \log T}\right) + \log(\frac{3C_1}{C_2})}{\frac{\log \log T}{\log \log \log T - \log C \log \log \log T}}$$

$$\leq 3\alpha,$$

for sufficiently large $T$. Let $\tau'_1, \ldots, \tau'_{\frac{\log \log T}{\log \frac{1}{3\alpha}}}$ be the update time defined in Remark 3 where $\alpha$ is replaced by $3\alpha$. Then from Theorem 3 we have $\tau'_{\frac{\log \log T}{\log \frac{1}{3\alpha}}} = \frac{T}{2}$. In addition, note that $\tau_{i+1} = 2^{-i} \tau_i^{\hat\alpha} T^{\hat\beta} = 2^{-i} \tau_i^{\hat\alpha} T^{\frac{1-\hat\alpha}{1-\frac{1}{\log T}}}$ decreases with $\hat\alpha$. Combining with (20), we have $\tau_{i+i_{j_0}} \geq \tau'_i$ for $i \in \{1, \ldots, S - i_{j_0}\}$. Thus, we have

$$\tau_{i_{j_0} + \frac{\log \log T}{\log \frac{1}{3\alpha}}} \geq \tau'_{\frac{\log \log T}{\log \frac{1}{3\alpha}}} = \frac{T}{2}.$$

As a result, with probability at least $1 - \frac{1}{T} - \frac{1}{T^2}$, with at most

$$S \leq i_{j_0} + \frac{\log \log T}{\log \frac{1}{3\alpha}} = O(\frac{\log \log T}{\log \frac{1}{\alpha}})$$

number of updates, $\tau_{S+1}$ is at least $T$. $\qquad\square$

## F  PROOF OF THEOREM 6

*Proof.* Note that whether changing distribution $\mu_t$ at time $t$ can be adversarially determined with the knowledge of $\{\tau_s : s \in \{1, \ldots, S\}, \tau_s \leq t\}$, where $\tau_s, s \in \{1, \ldots, S\}$, are the model update time determined by the update rule $\Phi$. Hence, for $S \leq K$, the distribution change time can be selected such that $\kappa_i = \tau_s + 1$. Then, the total loss is $\Omega(T)$. In the following, it is assumed that $S \geq K + 1$.

We show that Algorithm 2 achieves risk at most $\text{Risk}_{S,T}(\mathsf{P}) = O(S \left(\frac{T}{K+1}\right)^{\frac{1}{\lfloor \frac{S}{K+1} \rfloor + 1}})$. Let $\hat\kappa_1 < \ldots < \hat\kappa_k$ be the time when the if condition in line 6 in Algorithm 2 is satisfied and let $\hat\kappa_0 = 0$. Let $E_\ell$ be the event where there exists a distribution change at time $\kappa \in \{\hat\kappa_{\ell-1} + 1, \ldots, \hat\kappa_\ell\}$. We first show that $\Pr(\cap_{\ell=1}^k E_\ell) \geq 1 - \frac{1}{T}$. Note that $|\{i : \hat\kappa_{k-1} < \tau_i \leq \hat\kappa_k\}| = j_k \geq 1$ where $j_k$ is the value of $j$ when $t = \hat\kappa_k$. Since $E_\ell^c$ implies that $\mu_t$ is constant for $\hat\kappa_{\ell-1} < t \leq \hat\kappa_\ell$, according to (5) we have

$$\mathbb{E}_{\mathcal{D}_{\tau_{s(t)}}} \mathbb{E}_{(X,Y) \sim \mu_t} \left[\ell(\hat{h}_{s(t)}(X), Y)\right] \leq \frac{C_0 \log T}{\tau_{s(t)} - \kappa_{m(t)} + 1}$$

$$\leq \frac{C_0 \log T}{\tau_{s(t)} - \hat{\kappa}_k + 1}$$

$$\leq \frac{C_0 \log T}{\left(\frac{T}{K+1}\right)^{\frac{j_k}{\left\lfloor \frac{S}{K+1} \right\rfloor}}}$$

for $\tau_{s(t)} \leq t \leq \hat{\kappa}_\ell$, when $E_\ell^c$, $\ell \in \{1, \ldots, k\}$ occur. According to Bernstein inequallity (similar to Lemma 2), we have

$$\Pr(E_\ell^c) \leq exp(-\frac{\epsilon^2(\hat{\kappa}_\ell)}{\frac{2C_0 \log T(\hat{\kappa}_\ell - \tau_{i-1})}{\left(\frac{T}{K+1}\right)^{\left\lfloor \frac{S}{K+1} \right\rfloor + 1}} + \frac{2\epsilon(\hat{\kappa}_\ell)}{3}}) \leq \frac{1}{T^2}.$$

Therefore,

$$\Pr(\cap_{\ell=1}^k E_\ell) \geq (1 - \sum_{\ell=1}^k \Pr(E_\ell^c | \cap_{t=1}^T M_t))$$

$$\geq (1 - \frac{1}{T}) \tag{21}$$

Let $\Phi_1$ be the update rule given in Algorithm 2. Then,

$$\mathsf{Risk}_{S,T}(\Phi_1, \boldsymbol{\mu}) = \Pr(\cap_{\ell=1}^k E_\ell) \mathbb{E}_{\{(X_t, Y_t)\}_{t=1}^T \sim \boldsymbol{\mu}} \left[ \mathsf{Risk}(\Phi, \{(X_t, Y_t)\}_{t=1}^T) | \cap_{\ell=1}^k E_\ell \right]$$

$$+ \Pr(\cap_{\ell=1}^k E_\ell) \mathbb{E}_{\{(X_t, Y_t)\}_{t=1}^T \sim \boldsymbol{\mu}} \left[ \mathsf{Risk}(\Phi, \{(X_t, Y_t)\}_{t=1}^T) | \cap_{\ell=1}^k E_\ell \right]$$

$$\leq \mathbb{E}_{\{(X_t, Y_t)\}_{t=1}^T \sim \boldsymbol{\mu}} \left[ \mathsf{Risk}(\Phi, \{(X_t, Y_t)\}_{t=1}^T) | \cap_{\ell=1}^k E_\ell \right] + 2. \tag{22}$$

In the following, we provide an upper bound on the expectation $\mathbb{E}_{\{(X_t, Y_t)\}_{t=1}^T \sim \boldsymbol{\mu}} \left[ \mathsf{Risk}(\Phi, \{(X_t, Y_t)\}_{t=1}^T) | \cap_{\ell=1}^k E_\ell \right]$ conditioned on the event $\cap_{\ell=1}^k E_\ell$, i.e., there exist at least one distribution change between time $\hat{\kappa}_{\ell-1}$ and $\hat{\kappa}_\ell$ for $\ell \in \{1, \ldots, k\}$. The condition $\cap_{\ell=1}^k E_\ell$ implies that $k \leq K$. Note that from the if condition in line 8 in Algorithm 2, there are at most $\left\lfloor \frac{S}{K+1} \right\rfloor$ model updates between $\hat{\kappa}_{\ell-1}$ and $\hat{\kappa}_\ell$, $\ell \in \{1, \ldots, k+1\}$, where $\hat{\kappa}_{k+1} = T$. Therefore, there are at most $\left\lfloor \frac{S}{K+1} \right\rfloor (K+1) \leq S$ model updates in total. Denote

$$\tau_{\ell,1}, \ldots, \tau_{\ell,j_\ell} = \{\tau_i : i \in \{1, \ldots, S\}, \hat{\kappa}_{\ell-1} < \tau_i \leq \hat{\kappa}_\ell\}$$

as the collection of model update times between $\hat{\kappa}_{\ell-1}$ and $\hat{\kappa}_\ell$, $\ell \in \{1, \ldots, k+1\}$, where $\hat{\kappa}_{k+1} = T$. Then, since the if condition in line 6 is not satisfied at time $t \in \{\tau_{\ell,1}, \ldots, \tau_{\ell,j_k}, \tau_{\ell,j_k+1} - 1\}$ by definition, we have

$$\frac{\epsilon^2(t)}{\frac{2C_0 \log T(t - \tau_{\ell,j-1})}{\left(\frac{T}{K+1}\right)^{\left\lfloor \frac{S}{K+1} \right\rfloor + 1}} + \frac{2\epsilon(t)}{3}} < 2 \ln T$$

for $t = \tau_{\ell,j}$, $j \in \{1, \ldots, j_\ell\}$, and for $t = \tau_{\ell,j_\ell} - 1$, $\ell \in \{1, \ldots, k+1\}$, where $\tau_{\ell,0} = \hat{\kappa}_{\ell-1}$ and $\tau_{\ell,j_k+1} = \hat{\kappa}_\ell$. This implies that

$$\epsilon(\tau_{\ell,j}) \leq \frac{4 \ln T}{3} + \sqrt{\frac{16C_0 \log T \ln T(\tau_{\ell,j} - \tau_{\ell,j-1})}{\left(\frac{T}{K+1}\right)^{\frac{j}{\left\lfloor \frac{S}{K+1} \right\rfloor + 1}}}} + 1 \tag{23}$$

and that

$$\widehat{CL}(\tau_{\ell,j}) \leq \frac{C_0 \log T(\tau_{\ell,j} - \tau_{\ell,j-1})}{\left(\frac{T}{K+1}\right)^{\frac{j}{\left\lfloor \frac{S}{K+1} \right\rfloor + 1}}} + \frac{4 \ln T}{3} + \sqrt{\frac{16C_0 \log T \ln T(\tau_{\ell,j} - \tau_{\ell,j-1})}{\left(\frac{T}{K+1}\right)^{\frac{j}{\left\lfloor \frac{S}{K+1} \right\rfloor + 1}}}} + 1.$$

Therefore,

$$\mathbb{E}_{\{(X_t,Y_t)\}_{t=1}^T \sim \boldsymbol{\mu}}\left[\mathsf{Risk}(\Phi,\{(X_t,Y_t)\}_{t=1}^T)|\cap_{\ell=1}^k E_\ell\right] = \sum_{\ell=1}^{k+1}\sum_{j=1}^{j_\ell+1}\widehat{CL}(\tau_{\ell,j})$$

$$\leq \sum_{\ell=1}^{k+1}\sum_{j=1}^{j_\ell+1}\left(\mathbb{1}\left(j \leq \left\lfloor \frac{S}{K+1}\right\rfloor\right) + \mathbb{1}\left(j = \left\lfloor \frac{S}{K+1}\right\rfloor\right)\right)$$

$$\cdot \left(\frac{C_0\log T(\tau_{\ell,j}-\tau_{\ell,j-1})}{\left(\frac{T}{K+1}\right)^{\frac{j}{\lfloor\frac{S}{K+1}\rfloor+1}}} + \frac{4\ln T}{3} + \sqrt{\frac{16C_0\log T\ln T(\tau_{\ell,j}-\tau_{\ell,j-1})}{\left(\frac{T}{K+1}\right)^{\frac{j}{\lfloor\frac{S}{K+1}\rfloor+1}}}} + 1\right)$$

$$=O\left(S\ln T\left(\frac{T}{K+1}\right)^{\frac{j}{\lfloor\frac{S}{K+1}\rfloor+1}}\right) + \frac{C_0\log T(\sum_{\ell=1}^{k+1}\sum_{j=1}^{j_\ell+1}\mathbb{1}(j=\lfloor\frac{S}{K+1}\rfloor)(\tau_{\ell,j}-\tau_{\ell,j-1})}{\left(\frac{T}{K+1}\right)^{\frac{\lfloor\frac{S}{K+1}\rfloor}{\lfloor\frac{S}{K+1}\rfloor+1}}}$$

$$+\sum_{j=1}^{j_\ell+1}\mathbb{1}\left(j=\left\lfloor\frac{S}{K+1}\right\rfloor\right)\sqrt{\frac{16C_0\log T\ln T(\tau_{\ell,j}-\tau_{\ell,j-1})}{\left(\frac{T}{K+1}\right)^{\frac{j}{\lfloor\frac{S}{K+1}\rfloor+1}}}}$$

$$\leq O\left(S\ln T\left(\frac{T}{K+1}\right)^{\frac{1}{\lfloor\frac{S}{K+1}\rfloor+1}}\right) + \sum_{j=1}^{j_\ell+1}\mathbb{1}\left(j=\left\lfloor\frac{S}{K+1}\right\rfloor\right)\sqrt{\frac{16C_0\log T\ln T(\tau_{\ell,j}-\tau_{\ell,j-1})}{\left(\frac{T}{K+1}\right)^{\frac{j}{\lfloor\frac{S}{K+1}\rfloor+1}}}}$$

$$\leq O\left(S\ln T\left(\frac{T}{K+1}\right)^{\frac{1}{\lfloor\frac{S}{K+1}\rfloor+1}}\right) + \sqrt{16C_0\log T\ln T(K+1)\frac{\sum_{\ell=1}^{k+1}\sum_{j=1}^{j_\ell+1}\mathbb{1}(j=\lfloor\frac{S}{K+1}\rfloor)(\tau_{\ell,j}-\tau_{\ell,j-1})}{\left(\frac{T}{K+1}\right)^{\frac{\lfloor\frac{S}{K+1}\rfloor}{\lfloor\frac{S}{K+1}\rfloor+1}}}}$$

$$\leq O\left(S\ln T\left(\frac{T}{K+1}\right)^{\frac{1}{\lfloor\frac{S}{K+1}\rfloor+1}}\right) \tag{24}$$

which completes the proof of the upper bound for $\mathsf{Risk}_{S,T}(\mathsf{P})$.

Next, we deal with a lower bound showing that $\mathsf{Risk}_{S,T}(\mathsf{P}) = \Omega\left(S\ln T\left(\frac{T}{K+1}\right)^{\frac{1}{\lfloor\frac{S}{K+1}\rfloor+1}}\right)$ for

$$\mathbb{E}_{\mathcal{D}_{s(t)}}\mathbb{E}_{(X,Y)\sim\mu_t}\left[\ell(\hat{h}_{s(t)}(X),Y)\right] \geq O\left(\frac{1}{s(t)-\kappa_i+1}\right).$$

Let $\kappa_i = \frac{iT}{K+1}$ for $i \in \{1,\ldots,K\}$ and let $\tau_{k,j}$, $k \in \{1,\ldots,K+1\}$, $j \in \{1,\ldots,j_k\}$, be the model update time between $\kappa_{k-1}$ and $\kappa_k$ for any given update rule $\phi$, where $\kappa_0 = 1$ and $\kappa_{K+1} = T$. Then

$$\mathsf{Risk}_{S,T}(\mathsf{P}) \geq \mathbb{E}\left[\mathsf{Risk}(\Phi,\{(X_t,Y_t)\}_{t=1}^T)\right]$$

$$\geq \sum_{k=1}^{K+1}\sum_{j=1}^{j_k+1}\frac{(\tau_{k,j}-\kappa_{k-1})-(\tau_{k,j-1}-\kappa_{k-1})}{\tau_{k,j-1}-\kappa_{k-1}}$$

$$=\sum_{k=1}^{K+1}\sum_{j=1}^{j_k+1}\frac{\tau_{k,j}-\kappa_{k-1}}{\tau_{k,j-1}-\kappa_{k-1}} - \sum_{k=1}^{K+1}\sum_{j=1}^{j_k+1}1$$

$$\geq \sum_{k=1}^{K+1}\sum_{j=1}^{j_k+1}\frac{\tau_{k,j}-\kappa_{k-1}}{\tau_{k,j-1}-\kappa_{k-1}} - S - K$$

$$\geq \sum_{k=1}^{K+1}(j_k+1)\left(\frac{T}{K+1}\right)^{j_k+1}.$$

Note that for any $j_{k_1}-j_{k_2} \geq 2$, we have

$$(j_{k_1}+1)\left(\frac{T}{K+1}\right)^{j_{k_1}+1} + (j_{k_2}+1)\left(\frac{T}{K+1}\right)^{j_{k_2}+1} \geq (j_{k_1})\left(\frac{T}{K+1}\right)^{j_{k_1}} + (j_{k_2}+2)\left(\frac{T}{K+1}\right)^{j_{k_2}+2}$$

when $\frac{T}{K+1} \geq 2$. Hence, $\sum_{k=1}^{K+1}(j_k+1)\left(\frac{T}{K+1}\right)^{j_k+1}$ is minimized when $\max_{k_1,k_2 \in \{1,\dots,K+1\}} j_{k_1} - j_{k_2} = 1$. Therefore,

$$\text{Risk}_{S,T}(\mathsf{P}) \geq (K+1-m)\left(\left\lfloor \frac{S}{K+1} \right\rfloor + 1\right)\left(\frac{T}{K+1}\right)^{\frac{1}{\lfloor \frac{S}{K+1} \rfloor + 1}} + m\left(\left\lfloor \frac{S}{K+1} \right\rfloor + 2\right)\left(\frac{T}{K+1}\right)^{\frac{1}{\lfloor \frac{S}{K+1} \rfloor + 2}}$$

for any $S = \left\lfloor \frac{S}{K+1} \right\rfloor (K+1) + m$. Then, for any $S = \left\lfloor \frac{S}{K+1} \right\rfloor (K+1) + m$ for some $m \leq \frac{K+1}{2}$, we have

$$\text{Risk}_{S,T}(\mathsf{P}) \geq \frac{S}{2}\left(\frac{T}{K+1}\right)^{\frac{1}{\lfloor \frac{S}{K+1} \rfloor + 1}} = O\left(S\left(\frac{T}{K+1}\right)^{\frac{1}{\lfloor \frac{S}{K+1} \rfloor + 1}}\right).$$

For any $S = \left\lfloor \frac{S}{K+1} \right\rfloor (K+1) + m$ for some $m \geq \frac{K+1}{2}$, we have

$$\begin{aligned}
\text{Risk}_{S,T}(\mathsf{P}) &\geq \text{Risk}_{S+\frac{K+1}{2},T}(\mathsf{P}) \\
&\geq \frac{S + \frac{K+1}{2}}{2}\left(\frac{T}{K+1}\right)^{\frac{1}{\lfloor \frac{S}{K+1} \rfloor + 1}} \\
&= \Omega\left(S\left(\frac{T}{K+1}\right)^{\frac{1}{\lfloor \frac{S}{K+1} \rfloor + 1}}\right)
\end{aligned}$$

for $S \geq K+1$. Hence, $\text{Risk}_{S,T} = \Omega(S\left(\frac{T}{K+1}\right)^{\frac{1}{\lfloor \frac{S}{K+1} \rfloor + 1}})$ for any $S \geq K+1$. $\qquad\square$

## G  PROOF OF THEOREM 7

*Proof.* We construct two hypothesis classes such that under the same piecewise stationary data distribution and the oracle rule, the historical loss sequences under these two hypothesis classes and their respective choices of the ground truth function are hard to distinguish. On the other hand, the loss vectors under these two hypothesis classes follow the same behavior of loss vectors under realizable *i.i.d.* environment and piecewise stationary environment, respectively, and have expected risks far from each other. Then we use Le Cam's two point lemma to show that the existence of a universal risk-driven update algorithm $\mathcal{U}$ results in a contradiction.

To construct the first hypothesis class $\mathcal{H}_1$ based on Example 1 when $\alpha = \frac{1}{3}$, consider a partition of the indices $(l,j)$ of $8^l$ intervals $I_{l,j}$, $j \in \{1,\dots,8^l\}$, for each $l \in \mathbb{Z}_+$ into a set $\mathcal{P}_l = \{S_{l,1},\dots,S_{l,2^l+2l}\}$ of $2^l + 2l$ disjoint sets of intervals such that

$$|S_{l,v}| = \begin{cases} 1 & v = 1 \\ 2^{v-2} & 2 \leq v \leq 2l+1 \\ 4^l & 2l+2 \leq v \leq 2^l + 2l \end{cases}, \text{ and}$$

$$\bigcup_{v=1}^{2^l+2l} S_{l,v} = \{(l,1),\dots,(l,8^l)\} \tag{25}$$

Denote by $\mathcal{S} = (\mathcal{P}_l : l \in \mathbb{Z}_+)$ the collection of partitions $\mathcal{P}_l$ for all $l$ and denote $\boldsymbol{\sigma} = (\sigma_{l,j} : l \in \mathbb{Z}_+, j \in \{1,\dots,8^l\})$. Consider the hypothesis subclass

$$\mathcal{H}_{\boldsymbol{\sigma},\mathcal{S}} = \left\{\sum_{l=1}^{\infty}\sum_{v=1}^{2^l+2l} \theta_{l,v,\boldsymbol{\sigma},\mathcal{S}} G_{l,v,\boldsymbol{\sigma},\mathcal{S}}(x) : \theta_{l,v,\boldsymbol{\sigma},\mathcal{S}} \in \{-1,1\}\right\}$$

where

$$G_{l,v,\boldsymbol{\sigma},\mathcal{S}}(x) = \sum_{j=1}^{8^l} \mathbb{1}((l,j) \in S_{l,v})c_l\sigma_{l,j}g_{l,j}(x)$$

and $c_l = \frac{1}{2^{\frac{l}{2}}}$ for $l \in \mathbb{Z}_+$ and $v \in \{1, \ldots, 2^l + 2l\}$. Define the hypothesis class

$$\mathcal{H}_1 = \{h(x, \boldsymbol{\sigma}, \mathcal{S}) : h(x, \boldsymbol{\sigma}, \mathcal{S}) \in \mathcal{H}_{\boldsymbol{\sigma}, \mathcal{S}} \text{ for fixed } \boldsymbol{\sigma}, \mathcal{S}\},$$

where $x \in (0, 1)$, and $\boldsymbol{\sigma}$ and $\mathcal{S}$ take values over all possible choices of $\sigma_{l,j}$ and partition $\mathcal{S}_l$, respectively. Let the loss function be square loss and the oracle $\mathcal{O}^{\mathcal{H}_1}$ be the same[1] as in Example 1. More specifically, the oracle $\mathcal{O}^{\mathcal{H}_1}$ selects $\hat{h}^{\mathcal{H}_1}(x, \boldsymbol{\sigma}, \mathcal{S}) \in \mathcal{H}_1$ with the minimum $(\theta_{l,v,\boldsymbol{\sigma},\mathcal{S}} : l \in \mathbb{Z}_+, v \in \{1, \ldots, 2^l + 2l\})$ in lexicographic order such that the worst-case risk is minimized for every $\boldsymbol{\sigma}$ and $\mathcal{S}$.

The second hypothesis class is simply the hypothesis class $\mathcal{H}$ in Example 1 with dummy parameters $\boldsymbol{\sigma}$ and $\mathcal{S}$

$$\mathcal{H}_2 = \left\{ h(x, \boldsymbol{\sigma}, \mathcal{S}) : h(x, \boldsymbol{\sigma}, \mathcal{S}) = \sum_{l=1}^{\infty} \sum_{j=1}^{8^l} c_l \theta_{l,j} g_{l,j}(x), \theta_{l,j} \in \{-1, 1\} \text{for any } \boldsymbol{\sigma}, \mathcal{S} \right\}.$$

Similarly, the loss is square loss and the oracle $\mathcal{O}^{\mathcal{H}_2}$ returns $\hat{h}^{\mathcal{H}_2}(x, \boldsymbol{\sigma}, \mathcal{S}) \in \mathcal{H}_2$ with the minimum $(\theta_{l,v,\boldsymbol{\sigma},\mathcal{S}} : l \in \mathbb{Z}_+, v \in \{1, \ldots, 2^l + 2l\})$ in lexicographic order such that the worst-case risk is minimized for every $\boldsymbol{\sigma}$ and $\mathcal{S}$.

Consider the following piecewise stationary distribution with $K = 1$ distribution change. The initial distribution is given by $X_t = (U_t, \boldsymbol{\sigma}_0, \mathcal{S}_0) \sim \mu_0 = (U(0, 1), \boldsymbol{\sigma}_0, \mathcal{S}_0)$, where $\boldsymbol{\sigma}_0$ is all 1's vector and $\mathcal{S}_0 = \{\mathcal{P}_{l,0} : l \in \mathbb{Z}_+\}$ is any partition such that $\mathcal{P}_{l,0} = \{S_{l,1,0}, \ldots, S_{l,2^l+2l,0}\}$ satisfies (25).

Consider running a universal algorithm $\mathcal{U}$ on $\mathcal{H}_2$ with update budget $S = 2$ with initial data distribution $\mu_0$ and ground truth function $h^{\mathcal{H}_2}$. Upon time $t = \tau_1$ when $\mathcal{U}$ decides to update the model, the data distribution changes to $\mu_1$ defined in the following way. Let $\hat{h}_{\tau_1}^{\mathcal{H}_2}(x) = \sum_{l=1}^{\infty} \sum_{j=1}^{8^l} c_l \hat{\theta}_{l,j} g_{l,j}(x) \in \mathcal{H}_2$ be the output of the oracle $\mathcal{O}^{\mathcal{H}_2}(\{X_t, h^{\mathcal{H}_2}(X_t)\}_{t=1}^{\tau_1})$. In addition, let $h^{\mathcal{H}_2}(x, \boldsymbol{\sigma}, \mathcal{S}) = \sum_{l=1}^{\infty} \sum_{j=1}^{8^l} c_l \theta_{l,j} g_{l,j}(x)$. Denote the vector $\hat{\boldsymbol{\theta}} = (\theta_{l,j} : l \in \mathbb{Z}_+, j \in \{1, \ldots, 8^l\})$. For each $l \in \mathbb{Z}_+$ and $i \in \{0, \ldots, S\}$, let $V_l = \{(l, j) : \theta_{l,j} \neq \hat{\theta}_{l,j}, j \in \{1, \ldots, 8^l\}\}$ be the set of indices where there is a mismatch between $\theta_{l,j}, l \in \mathbb{Z}_+, j \in \{1, \ldots, 8^l\}$, for $h$ and $\hat{\theta}_{l,j,i}$ for $\hat{h}_{\tau_1}^{\mathcal{H}_2}$.

Let $|V_l| \bmod 4^l = \sum_{u=1}^{d_l} 2^{b_{l,u}}$, $|V_l| < 8^l$, be the binary representation of the remainder of $|V_l|$ divided by $4^l$ such that $b_{l,1} < \ldots < b_{l,d_l}$. For $|V_l| \bmod 4^l = 0$ and $|V_l| < 8^l$, let $d_l = 0$ and for $|V_l| = 8^l$, let $d_l = 2l + 1$ and

$$b_{l,u} = \begin{cases} 0 & \text{when } u = 1 \\ u - 2 & \text{when } 2 \le u \le 2l + 1. \end{cases}$$

Construct the partition $\mathcal{S}_1 = (\mathcal{P}_{l,1} : l \in \mathbb{Z}_+)$ such that $\mathcal{P}_{l,1} = \{S_{l,1,1}, \ldots, S_{l,2^l+2l,1}\}$ satisfies (25) for $l \in \mathbb{Z}_+$. Moreover, $\{S_{l,b_{l,u}+2,1} : u \in \{1, \ldots, d_l\}\} \cup \{S_{l,v,1} : v \in \{2l+2, \ldots, 2l + \lfloor \frac{|V_l|}{4^l} \rfloor + 1\}\}$ is a partition of $V_l$, i.e., $\mathcal{P}_{l,1} \backslash (\{S_{l,b_{l,u}+2,1} : u \in \{1, \ldots, d_l\}\} \cup \{S_{l,v,1} : v \in \{2l+2, \ldots, 2l + \lfloor \frac{|V_l|}{4^l} \rfloor + 1\}\})$ is a partition of $\{(l, j) : j \in \{1, \ldots, 8^l\}\} \backslash V_l$, for $l \in \mathbb{Z}_+$.

The distribution $\mu_2$ is given by $X_t = (U_t, \boldsymbol{\sigma}_1, \mathcal{S}_1) \sim \mu = (U(0, 1), \boldsymbol{\sigma}_1, \mathcal{S}_1)$, where $\boldsymbol{\sigma}_1 = -\hat{\boldsymbol{\theta}}$ flips every entry in $\hat{\boldsymbol{\theta}}$. Therefore, the data $X_t$ follows $\mu_1$ for $1 \le t \le \tau_1$ and $\mu_2$ for $\tau_1 < t \le T$.

Choose a ground truth function $h^{\mathcal{H}_2} \in \mathcal{H}_2$ such that the corresponding $\theta_{1,j} = 1$ for $j \in \{1, \ldots, 8\}$. Then, choose a ground truth function $h^{\mathcal{H}_1} \in \mathcal{H}_1$ such that

$$h^{\mathcal{H}_2}(X_t) = h^{\mathcal{H}_1}(X_t) \tag{26}$$

for $t \in \{1, \ldots, \}$. Note that this can be achieved by selecting

$$\theta_{l,v,-\hat{\boldsymbol{\theta}},\mathcal{S}_1} = \begin{cases} 1 & \text{when } v \in \{b_{l,u} + 2 : u \in \{1, \ldots, d_l\}\} \cup \{2l+2, \ldots, 2l + \lfloor \frac{|V_l|}{4^l} \rfloor + 1\} \\ -1 & \text{when } v \in \{1, \ldots, 2^l + 2l\} \backslash (\{b_{l,u} + 2 : u \in \{1, \ldots, d_l\}\} \cup \{2l+2, \ldots, 2l + \lfloor \frac{|V_l|}{4^l} \rfloor + 1\}) \end{cases}$$

---

[1]Note that the assumption on $\mathcal{O}^{\mathcal{H}_1}$ and $\mathcal{O}^{\mathcal{H}_2}$ can be removed by adding an extra vector $\boldsymbol{\sigma}_2$ in the function input in $\mathcal{H}_1$ and $\mathcal{H}_2$, respectively.

for $l \in \mathbb{Z}_+$. This follows from the definition of $G_{l,v,\hat{\sigma},\mathcal{S}}$, the fact that $\theta_{l,j} = -\hat{\theta}_{l,j}$ for $(l,j) \in V_l$, and the fact that $\{S_{l,b_{l,u}+2,1} : u \in \{1, \ldots, d_l\}\} \cup \{S_{l,v,1} : v \in \{2l+2, \ldots, 2l + \lfloor \frac{|V_l|}{4^l} \rfloor + 1\}\}$ is a partition of $V_l$.

Next, we show that

$$\hat{h}_{\tau_i}^{\mathcal{H}_1}(X_t) = \hat{h}_{\tau_i}^{\mathcal{H}_2}(X_t) \tag{27}$$

for $\tau_i < t \leq \tau_{i+1}$, $i \in \{0, 1\}$, where $\tau_0 = 0$ and $\tau_2$ is the time of the second model update decided by $\mathcal{U}$. For $\tau_0 < t \leq \tau_1$, $\hat{h}^{\mathcal{H}_1}$ and $\hat{h}^{\mathcal{H}_2}$ are initial functions returned by $\mathcal{O}^{\mathcal{H}_1}$ and $\mathcal{O}^{\mathcal{H}_2}$, respectively, where the parameters $\theta_{l,v,\sigma,\mathcal{S}} = -1$ in $\mathcal{H}_1$ and $\theta_{l,j} = -1$ in $\mathcal{H}_2$. Since the parameter $\sigma_0$ is all one vector for $\tau_0 < t \leq \tau_1$, we have

$$\hat{h}_{\tau_i}^{\mathcal{H}_1}(X_t) = \hat{h}_{\tau_i}^{\mathcal{H}_2}(X_t) = -\sum_{l=1}^{\infty} \sum_{j=1}^{8^l} c_l g_{l,j}(x).$$

We then show that (27) holds for $\tau_1 < t \leq \tau_2$. Note that $h^{\mathcal{H}_2}$ has corresponding $\theta_{1,j} = 1 \neq \hat{\theta}_{1,j}$ for all $j \in \{1, \ldots, 8\}$. As shown in the proof of Proposition 2, $\hat{\theta}_{1,j}$ can be corrected whenever there exists $U_t \in I_{1,j}$ for some $t \leq \tau_1$. Hence, the updated function $\hat{\mathcal{H}}_{2\tau_1}$ at time $t = \tau_1$ has a different set of $\theta$ compared to the initial all $-1$ vector. This implies that the updated function $\hat{h}_{\tau_1}^{\mathcal{H}_1}$ does not update the initial parameters $\theta_{l,v,-\hat{\theta},\mathcal{S}_1} = -1$ for $l \in \mathbb{Z}_+$ and $v \in \{1, \ldots, 2l+2^l\}$. Hence, we have

$$\hat{h}_{\tau_1}^{\mathcal{H}_1}(x, -\hat{\theta}, \mathcal{S}_1) = -\sum_{l=1}^{\infty} \sum_{v=1}^{2^l+2l} G_{l,v,-\hat{\theta},\mathcal{S}}(x) = \sum_{l=1}^{\infty} \sum_{j=1}^{8^l} c_l \theta_{l,j} g_{l,j}(x) = \hat{h}_{\tau_1}^{\mathcal{H}_2}(x, -\hat{\theta}, \mathcal{S}_1),$$

and thus (27) for $\tau_1 < t \leq \tau_2$.

Note that (26) and (27) imply that the loss vector $\ell^{\tau_2} = (\ell(X_t, h(X_t)) : t \leq \tau_2)$ is the same when the algorithm $\mathcal{U}$ is run on hypothesis classes $\mathcal{H}_1$ and $\mathcal{H}_2$ with oracles $\mathcal{O}^{\mathcal{H}_1}$ and $\mathcal{O}^{\mathcal{H}_2}$, respectively. Thus, by the Le Cam's two point lemma, any hypothesis class estimator $\psi : (0,1)^* \to \{\mathcal{H}_1, \mathcal{H}_2\}$ that takes the loss vector $\ell^{\tau_2}$ ($\tau_2$ can be decided by running $\mathcal{U}$) as input and returns the correct hypothesis class $\mathcal{H}_1$ or $\mathcal{H}_2$ that $\mathcal{U}$ runs on, has worst-case correct probability at most

$$\min \{\Pr(\psi(\ell_2^\tau)|\mathcal{U} \text{ runs on } \mathcal{H}_1) = \mathcal{H}_1), \Pr(\psi(\ell_2^\tau)|\mathcal{U} \text{ runs on } \mathcal{H}_2) = \mathcal{H}_2)\}$$

$$\leq 1 - \frac{1 - TV(\Pr(\ell^{\tau_2}|\mathcal{U} \text{ runs on } \mathcal{H}_1), \Pr(\ell^{\tau_2}|\mathcal{U} \text{ runs on } \mathcal{H}_2))}{2} = \frac{1}{2}$$

To derive a contradiction, in the following, we show that $\mathcal{U}$ results in a estimator $\psi$ with worst case correct probability at least $\frac{2}{3}$, i.e.,

$$\min \{\Pr(\psi(\ell_2^\tau)|\mathcal{U} \text{ runs on } \mathcal{H}_1) = \mathcal{H}_1), \Pr(\psi(\ell_2^\tau)|\mathcal{U} \text{ runs on } \mathcal{H}_2) = \mathcal{H}_2)\} \geq \frac{2}{3} \tag{28}$$

To this end, we arbitrarily pick a ground truth function $h^{\mathcal{H}_2} \in \mathcal{H}_2$ satisfying $\theta_{1,j} = 1$ for $j \in \{1, \ldots, 8\}$ and $\sum_{j=1}^{8^l} \theta_{l,j} = 0$, $l > 1$, i.e., half of the $\theta_{l,j}$'s, $j \in \{1, \ldots, 8^l\}$ take value in $1$ or $-1$, respectively, for each $l > 1$. Then, we pick a ground truth function $h^{\mathcal{H}_1} \in \mathcal{H}_1$ satisfying (26). The following lemma gives bounds on the expectation of the loss $\ell(\hat{h}(X_t), h(X_t))$, when $\mathcal{U}$ runs on $\mathcal{H}_1$ and $\mathcal{H}_2$, respectively. In Appendix H we prove the following lemma.

**Lemma 3.** *For hypothesis class $\mathcal{H}_2$, square loss, oracle $\mathcal{O}^{\mathcal{H}_2}$, and data distribution $X_t \sim \mu_0$ for $\tau_0 < t \leq \tau_1$ and $X_t \sim \mu_1$ for $\tau_1 < t \leq T$, where $\tau_1$ can be determined by running $\mathcal{U}$, we have*

$$\mathbb{E}_{\{X_i, h^{\mathcal{H}_2}(X_i)\}_{i=1}^{t_1}, X_{t_2}} \left[\ell(\hat{h}_{t_1}^{\mathcal{H}_2}(X_{t_2}), h^{\mathcal{H}_2}(X_{t_2}))\right] = \begin{cases} \Omega(\frac{1}{t_1^{\frac{1}{3}}}), O(\frac{\log t_1}{t_1^{\frac{1}{3}}}) & 1 \leq t_1 < t_2 \leq T \\ \Theta(1) & 0 = t_1 < t_2 \leq T \end{cases} \tag{29}$$

*and*

$$
\mathbb{E}_{\{X_i, h^{\mathcal{H}_1}(X_i)\}_{i=1}^{t_1}, X_{t_2}} \left[ \ell(\hat{h}_{t_1}^{\mathcal{H}_1}(X_{t_2}), h^{\mathcal{H}_1}(X_{t_2})) \right] = \begin{cases} \min\left\{ O\left(\frac{\log \tau_1}{\tau_1^{\frac{1}{3}}}\right), O(\frac{1}{t_1-\tau_1}) \right\} & \tau_1 < t_1 < t_2 \leq T \\ \Omega(\frac{1}{\tau_1^{\frac{1}{3}}}), O\left(\frac{\log \tau_1}{\tau_1^{\frac{1}{3}}}\right) & \tau_1 = t_1 < t_2 \leq T \\ O(\frac{1}{t_1}) & 0 < t_1 < t_2 \leq \tau_1 \\ \Theta(1) & 0 = t_1 < t_2 \leq \tau_1 \end{cases}
$$
(30)

*for* $0 \leq t_1 < t_2 \leq \tau_1$ *or* $\tau_1 < t_1 < t_2 \leq \tau_2$.

Define the following hypothesis class estimator

$$
\psi(\ell_2^\tau) = \begin{cases} \mathcal{H}_1 & \text{if } \tau_2 \leq T^{\frac{5}{6}} \\ \mathcal{H}_2 & \text{if } \tau_2 > T^{\frac{5}{6}}. \end{cases}
$$

In the following, we show that the correct probability of $\psi(\ell_2^\tau)$ is at least $\frac{2}{3}$, when $\mathcal{U}$ runs on $\mathcal{H}_1$ and $\mathcal{H}_2$, respectively. When $\mathcal{U}$ runs on $\mathcal{H}_1$, from Lemma 3 and Theorem 6, the optimal algorithm $\mathcal{U}$ achieves at most $O(\sqrt{T}\log T)$ expected loss. Then since $\tau_1$ is a stopping time decided by $\ell_1^\tau$ we have

$$
\mathbb{E}[\sum_{t=1}^{\tau_1} \ell(\hat{h}_0^{\mathcal{H}_1}(X_t), h^{\mathcal{H}_1}(X_t))] = \mathbb{E}[\tau_1]\mathbb{E}[\ell(\hat{h}_0^{\mathcal{H}_1}(X_1), h^{\mathcal{H}_1}(X_1))]
$$

Then, we have $\mathbb{E}[\tau_1] \leq O(\sqrt{T}\log T)$ and thus by Markov inequality,

$$
\Pr(\tau_1 > T^{\frac{7}{12}}) \leq \frac{\log T}{T^{\frac{1}{12}}} \leq \frac{1}{6}.
$$

On the other hand, when $\tau_1 \leq T^{\frac{7}{12}}$, we have

$$
\mathbb{E}[\sum_{t=\tau_1+1}^{\tau_2} \ell(\hat{h}_{\tau_1}^{\mathcal{H}_1}(X_t), h^{\mathcal{H}_1}(X_t))] = \mathbb{E}[\tau_2 - \tau_1]\mathbb{E}[\ell(\hat{h}_{\tau_1}^{\mathcal{H}_1}(X_{\tau_1+1}), h^{\mathcal{H}_1}(X_{\tau_1+1}))]
$$

and then, $\mathbb{E}[\tau_2] \leq O(\frac{T^{\frac{1}{2}}\log T}{T^{\frac{1}{36}}}) = O(T^{\frac{25}{36}}\log T)$. Hence by Markov inequality,

$$
\Pr(\tau_2 > T^{\frac{5}{6}}|\tau_1 \leq T^{\frac{7}{12}}) \leq \frac{\log T}{T^{\frac{5}{36}}} \leq \frac{1}{6}.
$$

Therefore,

$$
\Pr(\tau_2 > T^{\frac{5}{6}}) \leq \Pr(\tau_1 > T^{\frac{7}{12}}) + \Pr(\tau_2 > T^{\frac{5}{6}}|\tau_1 \leq T^{\frac{7}{12}}) \leq \frac{1}{3}.
$$

We next consider the case when $\mathcal{U}$ runs on $\mathcal{H}_2$ and show that $\tau_2 \leq T^{\frac{5}{6}}$ with probability at most $\frac{1}{3}$. Suppose on the contrary, $\tau_2 \leq T^{\frac{5}{6}}$ with probability at least $\frac{1}{3}$. Then we have

$$
\mathbb{E}[\sum_{t=\tau_2+1}^{T} \ell(\hat{h}_{\tau_2}^{\mathcal{H}_2}(X_t), h^{\mathcal{H}_2}(X_t))] = \mathbb{E}[T - \tau_2]\mathbb{E}[\ell(\hat{h}_{\tau_2}^{\mathcal{H}_2}(X_{\tau_2+1}), h^{\mathcal{H}_2}(X_{\tau_2+1}))]
$$

$$
\geq \Pr(\tau_2 \leq T^{\frac{5}{6}})\mathbb{E}[T - \tau_2|\tau_2 \leq T^{\frac{5}{6}}]\mathbb{E}[\ell(\hat{h}_{\tau_2}^{\mathcal{H}_2}(X_{\tau_2+1}), h^{\mathcal{H}_2}(X_{\tau_2+1}))|\tau_2 \leq T^{\frac{5}{6}}]
$$

$$
\overset{(a)}{\geq} \frac{T - T^{\frac{5}{6}}}{3}\Omega(\frac{1}{T^{\frac{5}{18}}})
$$

$$
= \Omega(T^{\frac{13}{18}}),
$$

where $(a)$ follows from (29). On the other hand, from Lemma 3 and Theorem 3[2], the optimal algorithm $\mathcal{U}$ achieves at most $O(T^{\frac{9}{13}})$ expected loss, which is a contradiction. Therefore, the probability that $\psi(\ell_2^\tau)$ makes a wrong estimation is at most $\frac{1}{3}$, which implies (28). This completes the proof. $\square$

---

[2]Note that here $(\mathsf{P}, \mathcal{H}_2, \mathcal{O}^{\mathcal{H}_2})$ is not a realizable *i.i.d.* learning environment. However, the learning curve $L(t)$ in the learning environment $(\mathsf{P}, \mathcal{H}_2, \mathcal{O}^{\mathcal{H}_2})$ follows the same behavior as that in an *i.i.d.* learning environment and thus Theorem 3 can be applied.

## H    PROOF OF LEMMA 3

*Proof.* By similar arguments as in the proof of Proposition 2, we have

$$\mathbb{E}_{X_{t_2}}[(h^{\mathcal{H}_2}(X_{t_2}) - \mathcal{O}^{\mathcal{H}_2}(\{X_i, h^{\mathcal{H}_2}(X_i)\}_{i=1}^{t_1}))^2]$$

$$= \begin{cases} \sum_{l=1}^{\infty} \sum_{j=1}^{8^l} \mathbb{1}(I_{l,j} \cap \{X_i\}_{i=1}^{t_1} = \emptyset) \frac{c_l^2}{8^l}(\sigma_{l,j}+1)^2 & t_1 > 0 \\ 2 & t_1 = 0 \end{cases}, \qquad (31)$$

for $t_2 > t_1$, since the value $h(X_t)$ for any $h \in \mathcal{H}_2$ does not depend on $(\boldsymbol{\sigma}, \mathcal{S})$ and the distribution of the loss vector $\ell^T$ is the same as that of the loss vector $\ell^T$ in Example 1. Note that

$$\mathbb{E}_{\{X_i, h^{\mathcal{H}_2}(X_i)\}_{i=1}^{t_1}}[\mathbb{1}(I_{l,j} \cap \{X_i\}_{i=1}^{t_1} = \emptyset)] = \left(1 - \frac{1}{8^l}\right)^{t_1}.$$

Combining with (31), we have

$$\mathbb{E}_{\{X_i, h^{\mathcal{H}_2}(X_i)\}_{i=1}^{t_1}, X_{t_2}}[(h^{\mathcal{H}_2}(X_{t_2}) - \mathcal{O}^{\mathcal{H}_2}(\{X_i, h^{\mathcal{H}_2}(X_i)\}_{i=1}^{t_1}))^2]$$

$$= \sum_{l=1}^{\infty} \frac{4}{2^l} \left(1 - \frac{1}{8^l}\right)^{t_1}$$

$$\leq \sum_{l=1}^{\frac{\ln t_1 - \ln \ln t_1}{\ln 8}} \frac{4}{2^l} \left(1 - \frac{1}{8^l}\right)^{t_1} + \sum_{l=\frac{\ln t_1 - \ln \ln t_1}{\ln 8}+1}^{\infty} \frac{4}{2^l} \left(1 - \frac{1}{8^l}\right)^{t_1}$$

$$\leq \frac{2^{\left(\frac{\ln t_1 - \ln \ln t_1}{\ln 8}\right)}}{t_1} + \frac{4 \ln^{\frac{1}{3}} t_1}{t_1^{\frac{1}{3}}} \leq O(\frac{\log t_1}{t_1^{\frac{1}{3}}}),$$

and

$$\mathbb{E}_{\{X_i, h^{\mathcal{H}_2}(X_i)\}_{i=1}^{t_1}, X_{t_2}}[(h^{\mathcal{H}_2}(X_{t_2}) - \mathcal{O}^{\mathcal{H}_2}(\{X_i, h^{\mathcal{H}_2}(X_i)\}_{i=1}^{t_1}))^2] \geq \frac{2}{e t_1^{\frac{1}{3}}} = O(\frac{1}{e t_1^{\frac{1}{3}}})$$

Therefore, (29) holds. Similarly, we have

$$\mathbb{E}_{X_{t_2}}[(h^{\mathcal{H}_1}(X_{t_2}) - \mathcal{O}^{\mathcal{H}_1}(\{X_i, h^{\mathcal{H}_1}(X_i)\}_{i=1}^{t_1}))^2]$$

$$= \begin{cases} \sum_{l=1}^{\infty} \sum_{v=1}^{2^l+2l} \mathbb{1}(\cup_{(l,j)\in S_{l,v,0}} I_{l,j} \cap \{X_i\}_{i=1}^{t_1} = \emptyset) \frac{|S_{l,v,0}|c_l^2}{8^l}(\theta_{l,v,\boldsymbol{\sigma}_0, \mathcal{S}_0}+1)^2 & 0 < t_1 < t_2 \leq \tau_1 \\ \sum_{l=1}^{\infty} \sum_{v=1}^{2^l+2l} \mathbb{1}(\cup_{(l,j)\in S_{l,v,1}} I_{l,j} \cap \{X_i\}_{i=1}^{t_1} = \emptyset) \frac{|S_{l,v,1}|c_l^2}{8^l}(\theta_{l,v,\boldsymbol{\sigma}_1, \mathcal{S}_1} - \hat{\theta}_{l,v,\boldsymbol{\sigma}_1, \mathcal{S}_1})^2 & \tau_1 \leq t_1 < t_2 \leq T \\ 2 & t_1 = 0 \end{cases},$$

$$(32)$$

Since

$$\mathbb{E}_{\{X_i, h^{\mathcal{H}_1}(X_i)\}_{i=1}^{t_1}}[\mathbb{1}(\cup_{(l,j)\in S_{l,v,u}} I_{l,j} \cap \{X_i\}_{i=1}^{t_1} = \emptyset)] = \left(1 - \frac{|S_{l,v,u}|}{8^l}\right)^{t_1},$$

we have

$$\mathbb{E}_{\{X_i, h^{\mathcal{H}_1}(X_i)\}_{i=1}^{t_1}, X_{t_2}}[(h^{\mathcal{H}_1}(X_{t_2}) - \mathcal{O}^{\mathcal{H}_1}(\{X_i, h^{\mathcal{H}_1}(X_i)\}_{i=1}^{t_1}))^2]$$

$$= \sum_{l=1}^{\infty} \sum_{v=1}^{2^l+2l} \left(1 - \frac{|S_{l,v,u}|}{8^l}\right)^{t_1} \frac{|S_{l,v,1}|c_l^2}{8^l}(\theta_{l,v,\boldsymbol{\sigma}_1, \mathcal{S}_1} - \hat{\theta}_{l,v,\boldsymbol{\sigma}_1, \mathcal{S}_1})^2$$

$$\leq \sum_{l=1}^{\infty} \sum_{v=1}^{2l+1} \left(1 - \frac{|S_{l,v,u}|}{8^l}\right)^{t_1} \frac{4|S_{l,v,1}|c_l^2}{8^l} + \sum_{l=1}^{\infty} \sum_{v=2l+2}^{2^l+2l} \left(1 - \frac{|S_{l,v,u}|}{8^l}\right)^{t_1} \frac{4|S_{l,v,1}|c_l^2}{8^l}$$

$$\leq \sum_{l=1}^{\infty} \sum_{v=1}^{2l+1} \left(1 - \frac{|S_{l,v,u}|}{8^l}\right)^{t_1} \frac{4|S_{l,v,1}|c_l^2}{8^l} + \sum_{l=1}^{\infty} \left(1 - \frac{1}{2^l}\right)^{t_1} \frac{4}{2^l}$$

$$\overset{(a)}{\leq} \sum_{l=1}^{\infty} (2l+1)\left(1 - \frac{1}{t_1+1}\right)^{t_1} \frac{c_l^2}{t_1+1} + \sum_{l=1}^{\infty} \left(1 - \frac{1}{2^l}\right)^{t_1} \frac{4}{2^l}$$

$$\leq O(\frac{1}{t_1}) + \sum_{l=1}^{\log t_1} \left(1 - \frac{1}{2^l}\right)^{t_1} \frac{4}{2^l} + \sum_{l=\log t_1+1}^{\infty} \left(1 - \frac{1}{2^l}\right)^{t_1} \frac{4}{2^l}$$

$$\overset{(b)}{\leq} O(\frac{1}{t_1}) + O(\frac{\log t_1}{t_1}) + O(\frac{1}{t_1}), \text{ and}$$

$$\mathbb{E}_{\{X_i, h^{\mathcal{H}_1}(X_i)\}_{i=1}^{t_1}, X_{t_2}}[(h^{\mathcal{H}_1}(X_{t_2}) - \mathcal{O}^{\mathcal{H}_1}(\{X_i, h^{\mathcal{H}_1}(X_i)\}_{i=1}^{t_1}))^2]$$

$$\geq \left(1 - \frac{1}{t_1}\right)^{t_1} \frac{1}{t_1}$$

for $0 < t_1 < t_2 \leq \tau_1$, where $(a)$ and $(b)$ follow from the fact that the function $(1 - x)^{t_1} x$ for $x \in (0, 1)$ is maximized at $x = \frac{1}{t_1+1}$. Similarly for $\tau_1 \leq t_1 < t_2 \leq T$, we have $\mathbb{E}_{\{X_i, h^{\mathcal{H}_1}(X_i)\}_{i=1}^{t_1}, X_{t_2}}[(h^{\mathcal{H}_1}(X_{t_2}) - \mathcal{O}^{\mathcal{H}_1}(\{X_i, h^{\mathcal{H}_1}(X_i)\}_{i=1}^{t_1}))^2] = O(1)$ for $t_1 = \tau_1$, $\mathbb{E}_{\{X_i, h^{\mathcal{H}_1}(X_i)\}_{i=1}^{t_1}, X_{t_2}}[(h^{\mathcal{H}_1}(X_{t_2}) - \mathcal{O}^{\mathcal{H}_1}(\{X_i, h^{\mathcal{H}_1}(X_i)\}_{i=1}^{t_1}))^2] \leq O(\frac{\log(t_2-t_1)}{t_2-t_1})$, and $\mathbb{E}_{\{X_i, h^{\mathcal{H}_1}(X_i)\}_{i=1}^{t_1}, X_{t_2}}[(h^{\mathcal{H}_1}(X_{t_2}) - \mathcal{O}^{\mathcal{H}_1}(\{X_i, h^{\mathcal{H}_1}(X_i)\}_{i=1}^{t_1}))^2] \geq O(\frac{1}{t_2-t_1})$ for $t_1 > \tau_1$. Finally, from (equation 27), (equation 26), and (equation 29) we have that $\mathbb{E}_{\{X_i, h^{\mathcal{H}_1}(X_i)\}_{i=1}^{t_1}, X_{t_2}}[(h^{\mathcal{H}_1}(X_{t_2}) - \mathcal{O}^{\mathcal{H}_1}(\{X_i, h(X_i)\}_{i=1}^{t_1}))^2] \leq \frac{\log \tau_1}{\tau_1^{\frac{1}{3}}}$ for $t_1 \geq \tau_1$, and $\mathbb{E}_{\{X_i, h^{\mathcal{H}_1}(X_i)\}_{i=1}^{t_1}, X_{t_2}}[(h^{\mathcal{H}_1}(X_{t_2}) - \mathcal{O}^{\mathcal{H}_1}(\{X_i, h(X_i)\}_{i=1}^{t_1}))^2] \geq \frac{1}{\tau_1^{\frac{1}{3}}}$ for $t_1 = \tau_1$ Therefore, we have (30). $\qquad \square$

