# OpenReview forum: "How to Retrain Online Models Optimally with Few Updates"
_ICLR.cc/2026/Conference — Submitted to ICLR 2026_

### Official Review · Reviewer_ZzpT · 2025-11-01

**Soundness:** 3
**Presentation:** 4
**Contribution:** 3
**Rating:** 8
**Confidence:** 2

**Summary:**

This paper studies a central question in online learning: how frequently does one need to retrain an online model to maintain near-optimal cumulative predictive performance. Under mild assumptions, the authors establish two notable results:
1. With non-increasing learning curve, $\log T$ updates are sufficient for a schedule optimal up to constant factor
2. With power-law learning curve with $\alpha < 1$, only $\log \log T$ updates are required. It's noteworthy that $\alpha < 1$ aligns with empirical scaling-law observations in large language models, making the result particularly relevant.

These results are further strengthened by
- An update schedule for case (2) that does not require knowing $\alpha$ in advance
- An extension to a piecewise-i.i.d. setting to address distribution shift
- A no-free-lunch result showing that optimal update rules are impossible without assumptions
- An empirical demonstration of (2).

**Strengths:**

- The authors study a timely and practically important question through a rigorous theoretical lens. The assumptions align well with empirical results, making the work highly relevant to current practice.
- The theoretical contributions are strong and surprising. These results are well-complemented by $\alpha$-agnostic update schedules, no-free-lunch theorem, and extension to distribution shift setting.
- The writing and organization are clear, polished, and accessible
- The empirical results nicely corroborates the theoretical predictions that only doubly-exponentially fewer updates are required.

**Weaknesses:**

- The author does not provide empirical results for Eq. 7, which should be quite easy to collect.

**Questions:**

- On line 92, the authors point to Algorithm 6. Should this refer to Algorithm 2 instead?
- Could the authors provide experimental results in a language-model setting? Given the strong motivation from LLM scaling-law behavior, such experiments would significantly enhance the practical relevance and appeal of the work

---

> ### Author Response · Authors · 2025-11-15
>
> We thank the reviewer for the highly positive and encouraging assessment of our work. We appreciate the recognition of both the theoretical depth and the practical relevance of our results. We address the main questions below.
>
> **Empirical results and clarifications**
> We thank the reviewer for catching the typo—“Algorithm 6” should indeed refer to *Algorithm 2*, and we will correct this in the revision. We also agree that adding experimental results for Eq. 7 is straightforward and will include them in the updated version.
>
> Regarding the suggested language-model experiments, we agree that this is an exciting and natural extension. However, our current exposition focuses primarily on the conceptual and theoretical development of optimal retraining in online learning. The experiments are mainly intended as proof-of-concept validation. We believe such extensions would be an interesting and valuable direction for future work.

---

### Official Review · Reviewer_SH8c · 2025-11-01

**Soundness:** 3
**Presentation:** 3
**Contribution:** 2
**Rating:** 2
**Confidence:** 3

**Summary:**

This paper studies the problem of determining the optimal retraining schedule for online learning models under limited update budgets. In a typical online learning setup, the model observes data sequentially and incurs prediction loss at each round. Retraining updates improve performance but are computationally expensive. The authors formalize a setting where only $S$ retraining events are allowed over a time horizon of $T$ steps and analyze the trade-off between retraining frequency and cumulative predictive risk.

The paper focuses primarily on the realizable i.i.d. and piecewise-stationary environments. Under these settings, the authors characterize optimal retraining intervals in terms of the learning curve decay rate. They show that for non-increasing learning curves, only $O(\log T)$ retraining events are sufficient to match the performance of continuous retraining within a constant factor. For power-law learning curves $L(t) \sim t^{-\alpha}$, they further prove that only $O(\log \log T)$ retrainings are needed to achieve near-optimal risk. The paper also proposes adaptive algorithms that do not require prior knowledge of the learning curve exponent and extends the analysis to piecewise-stationary distributions. Theoretical guarantees are supported by small-scale experiments.

**Strengths:**

1. The paper provides rigorous mathematical analysis and establishes sharp retraining-frequency guarantees.
2. The results offer interesting theoretical insights regarding the relationship between learning curves and update schedules.
3. The proposed algorithms are backed by solid proofs and demonstrate provable adaptivity to unknown parameters.

**Weaknesses:**

The primary concern lies in the core assumptions that motivate the problem. In practice, retraining is typically driven by distributional changes rather than simply the accumulation of more data from an unchanged distribution. However, the paper assumes i.i.d. or piecewise-stationary environments, where retraining due to distribution shift is either unnecessary (in the i.i.d. case) or explicitly handled by assumed segmentation (piecewise-stationary case). This disconnect raises concerns about the practical relevance of the problem formulation.

As a result, the work appears to build an elegant theoretical framework atop assumptions that do not match the settings where retraining is most critical in real deployments. Although the analysis is technically sound, the paper's motivation does not convincingly justify why optimizing retraining schedules under stationary or near-stationary distributions yields meaningful practical benefit.

**Questions:**

1. Can the authors clarify concrete real-world scenarios where retraining frequency matters under stationary or piecewise-stationary distributions?
2. In practice, how would the proposed methods generalize to environments where model drift is caused by gradual changes in the distribution rather than clean piecewise segments?

---

> ### Author Response · Authors · 2025-11-15
>
> We thank the reviewer for the detailed feedback and for recognizing the rigor and sharpness of our theoretical analysis. We respectfully disagree with the concern that our problem formulation lacks practical relevance, and we clarify below why our framework is both *novel* and *fundamental* to understanding retraining dynamics in modern online learning.
>
> **Novelty and scope of our contribution**
> To our knowledge, our work is the *first* to formally treat *retraining frequency* itself as an optimization objective within online learning. We know that while some prior work has studied online adaptation or learning rate schedules, none have provided theoretical characterizations of *how often* one should retrain when retraining is costly. Our main results (even in i.i.d. setting), such as the competitive optimality relative to full-update retraining, *universal* schedule for any non-increasing learning curve, the doubly-exponential improvement for power-law curves, and the impossibility theorems (Theorems 4 and 7), are all conceptually strong and far from trivial. Together, they establish a **new theoretical foundation** for reasoning about retraining under limited update budgets, independent of specific model architectures or learning algorithms.
>
> **Practical relevance beyond strict stationarity**
> The reviewer notes that retraining is often motivated by distribution shift rather than data accumulation. We fully agree, and our framework directly connects to this (see the discussion of piecewise-stationary data below). Even in stationary settings, retraining frequency remains a critical design choice in continual learning, where retraining can incur computational, stability, and adaptation costs independent of distributional changes. Our analysis provides general principles for when such retraining is *worthwhile*, offering a unifying theoretical foundation for more complex, non-stationary environments.
>
> **Piecewise-stationary and beyond**
> The piecewise-stationary case we study is *non-trivial*: the change points are **unknown** to the learner, and our adaptive algorithm (Algorithm 2) provably detects and responds to them based solely on risk feedback (i.e., our central concept of *risk-driven* updates). This already models realistic forms of abrupt drift without oracle knowledge of segmentation.  Moreover, we consider i.i.d. and piecewise-stationary cases because they are easier to illustrate our concepts and ideas and because they are fundamental to understand more complex data distributions.
>
> In fact,  our framework **can** naturally extend to *gradual or continuous* distribution shifts by interpreting the learning curve as a slowly varying function over time and allowing the risk-driven mechanism to adaptively trigger updates as the model’s performance degrades. As a specific example, consider the case where the data $X_t$ follows a distribution $\mu_t$ such that the total variation between $\mu_t$ and $\mu_{t-1}$ is upper bounded by some parameter $\gamma$, i.e., $|\mu_t-\mu_{t-1}|_{TV}\le \gamma$, for $t\in\{2,\ldots,T\}$. The following paper (Theorem 17 and Theorem 18)
>
> Peter L. Bartlett, “Learning with a Slowly Changing Distribution.”
> In Proceedings of the Fifth Annual Workshop on Computational Learning Theory (COLT ’92), pp. 243–252, 1992.
>
> showed that $L(t)\le O(\log t/t)$ for hypothesis set $\mathcal{H}$ with finite VC dimension, when $\gamma\le O(\frac{1}{t^3})$ (meaning that for $\gamma<1$, the learning rate in a short time window can be roughly bounded by learning rate for the i.i.d. case). To see how our adaptive algorithm (Algortihm 2) can be applied here, let $T_{\gamma}:=1/{\gamma}^3$. Then the gradual distribution changes across time horizon $T$ can be approximated by $T/T_{\gamma}$ piecewise stationary distributions. This implies that our Algorithm 2 will be able to achieve optimal loss with at most $O((T/T_{\gamma})\cdot\log T)$ model updates, which is much better than update at each time.
> The gradual data distribution case under different assumptions may require additional technical analysis that is worthy of an independent study. In fact, our formulation in Section 2 explicitly models the general data process though the notion of **learning environment**. In addition, we would like to note that our Theorem 7 hints that there does not exist a general algorithm that is optimal for all cases.
>
> Overall, we believe that adding new learning environments to the current exposition would primarily expand the *quantitative scope* of the paper (though technically interesting) without leading to any significant new qualitative insights.
> Our main contribution, however, lies in providing a **conceptual shift** that treats *retraining frequency* itself as an optimization objective within online learning. We believe this perspective is adequately addressed in the paper through a series of non-trivial theoretical results, both positive and negative.

---

### Official Review · Reviewer_h3gc · 2025-11-03

**Soundness:** 3
**Presentation:** 2
**Contribution:** 3
**Rating:** 6
**Confidence:** 4

**Summary:**

This paper addresses the problem of optimal retraining frequency for online models, balancing predictive performance against the high computational costs of updating.The authors demonstrate, within an i.i.d. realizable online learning framework, that full retraining at every step is unnecessary, as $O(\log T)$ updates can achieve near-optimal risk. Furthermore, the paper provides algorithms that achieve these guarantees, including adaptive methods for unknown learning curve parameters and extensions to piecewise-stationary environments.

**Strengths:**

This paper makes an interesting contribution to the literature on online learning and continual learning. The theoretical characterization of retraining frequency is a valuable result, establishing that near-optimal risk can be achieved with exponentially fewer updates (e.g., $O(\log T)$ or $O(\log \log T)$) than a full update schedule, depending on the learning curve's decay rate.

A particularly compelling aspect of this work is the development of "risk-driven" update rules. The idea of monitoring the empirical risk to dynamically trigger updates is an essential and practical concept. This approach is a key strength of the paper and warrants the emphasis the authors have placed on it.

**Weaknesses:**

While the paper's contributions are noteworthy, there are several areas where it could be significantly improved. The comments below are offered as constructive suggestions.

* **Clarity of Introduction:** The paper would benefit from a revised introduction. The motivation for the problem, as well as a clear overview of the paper's specific goals and structure, could be articulated more clearly. Highlighting the "risk-driven" update mechanism as a core conceptual contribution earlier in the paper could also strengthen its positioning.
* **Formulation of Optimality:** The current formulation of "optimality" is based on minimizing the *number* of updates (S) for a given time horizon T. A significant suggestion is to reformulate the problem to account for the *cost* of retraining. As the authors allude to, the cost of retraining is not constant; it typically grows with the amount of data (*t*). An optimal schedule should ideally be defined by a total computational budget, not just a fixed number of updates.
* **Extension on Cost-Performance Trade-off:** Following the previous point, a valuable extension would be to analyze the trade-off curve plotting total computational cost (derived from an optimal schedule) against predictive performance as the total budget is varied.
* **Reliance on Strong Assumptions:** The main analysis relies on strong assumptions about the learning environment (e.g., i.i.d. realizable data, known learning curve shapes). The paper then moves to settings where these assumptions are relaxed, but the analysis feels somewhat heuristic.
* **Lack of Empirical Validation:** The most significant weakness of the current submission is the near-total lack of empirical validation. The theoretical claims are interesting, but they require empirical support to be fully convincing.
    * In settings where the paper's assumptions *are* met, experiments should be provided to demonstrate that the proposed schedules (e.g., from Theorem 2 or 3) are indeed optimal in practice.
    * In settings where assumptions are *not* met (e.g., unknown environment parameters), the robustness of the proposed methods must be tested.
    * The two main algorithms proposed (the adaptive algorithm for unknown $\alpha$ mentioned in the text and Algorithm 2 for piecewise-stationary data) currently lack sufficient empirical backing.
* **Suggested Application:** A very interesting application and extension would be to apply these scheduling approaches to a language modeling task. Even a small-scale experiment, testing the cases with both known and estimated exponents, would significantly strengthen the paper's practical relevance.
* **Minor Points:** The authors might consider a more descriptive title. The final conclusion section also appears unfinished and requires revision.

**Questions:**

1.  **Regarding Theorem 3:** The theorem statement provides a risk bound, but it does not explicitly define the schedule used to achieve it. The proof appears to rely on a specific schedule construction. Could the authors clarify in the theorem statement that this bound is achieved *by* a specific schedule, or clarify if it holds more generally?
2.  **A question on notation:** In Algorithm 2, line 5, the update condition uses both `log T` and `ln T`. Is the use of the natural logarithm (`ln`) intentional, or is this a typographical inconsistency? A similar question stands if "Algorithm 1" (which was not visible in the provided PDF) has a similar inconsistency, as noted by the reviewer.

---

> ### Author Response · Authors · 2025-11-15
>
> We thank the reviewer for the thoughtful and detailed feedback, and for recognizing the novelty of our theoretical and conceptual contributions. We address the main points below.
>
> **Clarity of introduction and presentation**
> We appreciate this suggestion and will emphasize the *risk-driven* mechanism earlier in the introduction, and present it as a central conceptual contribution of the work.
>
> **Formulation of optimality and cost–performance trade-off**
> Indeed, our current formulation focuses primarily on minimizing the *number* of retrainings over a given horizon, abstracting each retraining as a single oracle call. This abstraction is central to our contribution, as it isolates the *frequency* of retraining as a core *primitive* independent of the oracle’s specific computational details. Nonetheless, our framework can naturally accommodate non-uniform or data-dependent retraining costs $c_t$ (e.g., reflecting the size of the data used) by imposing a budget constraint $\sum_t c_t \leq C$. We emphasize that *frequency* should remain the primary optimization objective, since directly optimizing total computational cost may lead to suboptimal (i.e., overly frequent) schedules even when they appear computationally cheaper. Note that our retraining scheme still achieves a total computational complexity of $O(T)$, *matching* naive incremental-update schemes (which lack formal predictive-performance guarantees for general classes and may require *exponentially* more retrainings). We refer the reviewer to our response to Reviewer VQJi for a more detailed discussion of a related concern.
>
> **Assumptions and extensions**
> We agree that the i.i.d. setting provides an idealized baseline. However, Theorems 5–7 and Algorithm 2 explicitly extend our results to piecewise-stationary environments, where we provide provable adaptive guarantees. We will clarify these assumptions and highlight their practical implications in the revised version. We also would like to point out that we can easily generalize Theorem 1 by dropping the i.i.d. assumption and replacing non-increasing of $L(t)$ by a weaker statement like non-increasing upper and lower bounds on $L(t)$ that are orderwise the same. More specifically, our strategy in Theorem 1 is optimal whenever $L_1(t)\le L(t)\le L_2(t)$ for non-increasing functions $L_1(t),L_2(t)$ satisfying $L_2(t)\le CL_1(t)$ for some  constant $C$. Furthermore, as elaborated in our detailed response to Reviewer SH8c, our framework can be naturally extended to settings with gradual or continuous distribution shifts. The intuition is that the learning curve can be viewed as a slowly varying function of time, and the risk-driven update mechanism will adaptively trigger updates whenever the model’s estimated performance begins to degrade. The technical foundation for this extension lies in our comprehensive analysis of both the i.i.d. and piecewise-stationary i.i.d. regimes presented in our paper. Not surprisingly, the simplest setting (e.g., the i.i.d. case) is also the most intellectually illuminating; extending the analysis to more general models typically requires additional technical machinery but does not yield fundamentally new qualitative insights.
>
> **Empirical validation and notation**
> We agree that our current experiments are limited (though our additional experiments exhibit similar phenomena on CIFAR-10). However, the main focus of this paper is conceptual and theoretical, we aim to shift the paradigm from simply minimizing regret or risk to incorporating *retraining frequency* as a primary optimization objective in online learining. The experiments are intended primarily as a proof of concept. We agree that extending to larger-scale experiments (e.g., on language models) would be a valuable future direction to explore.
>
> **Q1:** The risk bound in Theorem 3 is achieved by the schedule $\tau_i = T^{\frac{1 - \alpha^i}{1 - \alpha^{S+1}}}$, as detailed in the corresponding proof.
>
> **Q2:** We apologize for the confusion. We use $\log T$ to denote the logarithm with an unspecified base (i.e., the results hold for any constant base). However, in Algorithm 2, line 5, the base is specifically chosen as $e$ to match the constants in the Bernstein inequality. We will clarify this in the revision.

---

### Official Review · Reviewer_TiTP · 2025-11-03

**Soundness:** 4
**Presentation:** 3
**Contribution:** 2
**Rating:** 4
**Confidence:** 3

**Summary:**

This work investigates the need for model retrains over time through the lens of online learning.
Specifically, they show that schedules with $O(\log T)$ retrains are sufficient to be within a constant factor of the
optimal risk (i.e., mean loss over time). This result is based on clean connections to the ``doubling trick'' in
online learning. The authors give tighter risk bounds in the case of certain
sublinear learning curves (e.g., power law in Theorem 3). Finally, the authors extend their analysis somewhat to
piecewise-stationary distributions with distribution shift and give some numerical results (Figure 2).

**Strengths:**

- This work provides a precise characterization of the trade-off between retrain frequency and predictive risk.
- Tighter analysis for sublinear learning curves of the form $O(1/t)$ and $O(1/t^\alpha)$, i.e., Theorem 2 and Theorem 3.
- Starts to study aspects of distribution shift, which is the most practical case.
- Algorithm 1: Adaptive algorithm for power law learning curves if exponent $\alpha$ isn't known.
- Nice concluding no-free lunch theorem (Theorem 7), showing that some prior information is needed to be competitive.

**Weaknesses:**

- Limited discussion about how this retrain schedule framework and the results relate to previous work.
- While the theoretical framework and results are clean, this manuscript could benefit from strong experiments.

**Questions:**

**Questions**
- [152] As $t$ increases we have access to more data. How (if any) can this
  work generalize to non-uniform retrain costs? Intuitively, the first rounds
  could be cheaper retrains.

**Misc**
- [130] Suggestion: Missing reference for learning under nonstationarity: "Learning Rate Schedules in the Presence of Distribution Shift" [Fahrbach et al., ICML 2023]

---

> ### Author Response · Authors · 2025-11-15
>
> We thank the reviewer for the thoughtful feedback and for acknowledging our technical contributions. We address the main concerns below.
>
> **Generalization to non-uniform retrain costs**
> Indeed, our current formulation focuses primarily on minimizing the *number* of retrainings over a given horizon, abstracting each retraining as a single oracle call. This abstraction is central to our contribution, as it isolates the *frequency* of retraining as a core *primitive* independent of the oracle’s specific computational details. Nonetheless, our framework can naturally accommodate non-uniform or data-dependent retraining costs $c_t$ (e.g., reflecting the size of the data used) by imposing a budget constraint $\sum_t c_t \leq C$. We emphasize that *frequency* should remain the primary optimization objective, since directly optimizing total computational cost may lead to suboptimal (i.e., overly frequent) schedules even when they appear computationally cheaper. Note that our retraining scheme still achieves a total computational complexity of $O(T)$, *matching* naive incremental-update schemes (which lack formal predictive-performance guarantees for general classes and may require *exponentially* more retrainings). We refer the reviewer to our response to Reviewer VQJi for a more detailed discussion of a related concern.
>
> **Discussion of prior work**
> We thank the reviewer for suggesting relevant references (e.g., *Fahrbach et al., ICML 2023*) and will include them in the revision. To our knowledge, our work is the first to explicitly treat *retraining frequency* as an optimization objective within online learning. We would be happy to incorporate any additional related work the reviewer recommends.
>
> **Experiments**
> We agree that our current experiments are limited to MNIST with small neural networks (though our additional experiments also exhibit similar phenomena on CIFAR-10, and we will be happy to include in the revisions). However, our paper is primarily conceptual and theoretical, introducing a *new paradigm* for reasoning about retraining frequency in online learning. Our experiments are intended mainly as proofs of concept.

---

### Official Review · Reviewer_VQJi · 2025-11-10

**Soundness:** 2
**Presentation:** 2
**Contribution:** 2
**Rating:** 4
**Confidence:** 3

**Summary:**

This paper provides the first precise characterization of optimal retraining frequency for online models, demonstrating that retraining at every step is unnecessary. It proves that $O(\log T)$ updates are sufficient for general i.i.d. data, and a doubly-exponentially small $O(\log \log T)$ updates are sufficient when the learning curve follows a common power-law decay.

**Problem formulation**

The paper models online learning with a fixed retraining budget of $S$ oracle calls over a time horizon of $T$ steps. At each update time $\tau_s$, a predictor $h_s$ is trained, and this predictor is used for all predictions until the next update at $\tau_{s+1}$. The objective is to design the update schedule $\tau_1, \cdots, \tau_S$ to minimize the total cumulative risk, $Risk_{S,T} = \sum_{t=1}^{T}l(h_{s(t)}(X_{t}),Y_{t})$, where $s(t)$ is the index of the most recent update.

**Main results**

The main theoretical results show that for any non-increasing learning curve, $S=O(\log T)$ updates achieve a risk within a constant factor of the optimal full-update (S=T) risk. Furthermore, if the learning curve follows a power law $L(t) \sim 1/t^\alpha$ (with $\alpha < 1$), only $S=O(\log \log T)$ updates are needed to match the optimal full-update risk.

**Technique/algorithm**

The paper first proposes "fixed-design" schedules, such as a geometric "doubling trick" ($\tau_i = 2^{i-1}$), that depend on the general shape of the learning curve. To handle unknown environments, it develops "risk-driven" adaptive algorithms (like Algorithm 1) that monitor empirical losses to estimate the learning curve's parameters or detect distribution shifts (Algorithm 2), allowing the schedule to adapt and remain optimal.

**Experiment sumamry**

Experiments on the MNIST dataset validate the theory by comparing a full-update (single SGD step per round) baseline to the proposed $O(\log \log T)$ schedule. The results confirm that this "$\Phi$-schedule," using doubly-exponentially fewer updates, achieves a cumulative risk that is within a small constant factor ( e.g 3-4) of the baseline.

**Strengths:**

The paper formulates retraininig in the online learning framework and provides some theory.

**Weaknesses:**

The theory is largely expected from existing online learning technique.
Also the paper models retraining as a single "oracle call," which is experimentally treated as a full, computationally massive retrain on the entire data prefix, rather than a more practical incremental update.

**Questions:**

.

**Details Of Ethics Concerns:**

.

---

> ### Author Response · Authors · 2025-11-15
>
> We thank the reviewer for the thoughtful feedback and for summarizing our work accurately. We now address the main concerns below:
>
> **“The theory is largely expected from existing online learning techniques.”**
> To our knowledge, there is no prior literature that treats the *frequency* of retraining itself as a primary optimization objective. Our main conceptual contributions, such as the competitive optimality with respect to the full-update schedule, a *universal* rule for any non-increasing learning curve, the doubly-exponential improvement under power-law scaling, and several impossibility results (e.g., Theorems 4 and 7), are far from straightforward and, in several cases, quite **surprising**.
> We therefore believe that it would be unjustified to describe this collection of deep theoretical results as “expected.” If the reviewer is aware of any prior work that directly supports their perspective, we would be grateful if they could share those references for our consideration.
>
> **On the computational cost and oracle modeling:**
> Our primary focus in this paper is on the *frequency* of retraining, while we treat the cost of each retrain as a single oracle call. We emphasize that our framework is not constrained to any specific oracle; rather, our main theorems provide general *characterizations* of when to optimally invoke a **given** oracle based on its predictive performance (e.g., the learning curve). This abstraction is powerful, deliberate, and central to our contribution, as it provides a clean formulation that isolates the optimization over *frequency* from the internal implementation of the oracle.
>
> Importantly, retraining *frequency* captures not only computational cost but also instability, catastrophic forgetting, and user adaptation overheads that arise from continuous model deployment. Directly minimizing total computational cost may lead to suboptimal (i.e., frequent) update schedules, even if they appear computationally cheaper overall. In any case, *frequency* remains an important primitive. The fact that other factors may also matter does not undermine the fundamental nature of our contribution in this paper.
>
> Regarding the reviewer’s concern about the computational cost of full-prefix retraining, we note that our retraining scheme still achieves a total computational complexity of $O(T)$, *matching* the naive incremental update schemes (which have no formal predictive-performance guarantees for general classes and require *exponentially* more retrainings). Furthermore, our framework **does not** exclude the use of an incremental-update oracle (such as in Algorithm 2), provided that its predictive performance (e.g., its learning curve) satisfies the premises of our theorems.

---

### Meta-Review · Area_Chair_7ecY · 2026-01-06

**Summary:**

The reviewers identified the following strengths and concerns (S=strengths and C=concerns)

S1. Paper makes an interesting contribution to online/continual learning by characterizing the risk in terms of the retraining frequency and the decay rate.

S2. The paper leads to the idea of useful rules based on monitoring the empirical risk when deciding whether to perform model updates.

C1. The formation of optimality should take into account the cost of retraining.

C2. The analysis relies on strong assumptions (i.i.d. and piecewise stationary environments) that are unlikely to be satisfied in practice. This raises the question as to whether the theoretical framework fails to match the practical setting where retraining decisions are most critical.

C3. There is a lack of empirical validation. The paper does not experimentally confirm that the proposed schedules are optimal when assumptions are met and does not provide any experimental analysis to demonstrate robustness to violations of the assumptions.

While the reviewers all acknowledged positive aspects to the paper, there were several important criticisms raised. The authors did not make use of the opportunity to submit a revised version of the paper that would take into account and partially address the reviewers' concerns. Although it can be accepted that the request for empirical validation isn't easy to address during a short rebuttal period, the response did argue that assumptions could be relaxed and results extended to address the other concerns. The response reads as though these are relatively easy extensions/relaxations: "we can easily generalize Theorem 1 by dropping the i.i.d. assumption", "our framework can be naturally extended to settings with gradual or continuous distribution shifts". Given these statements in the response, providing a revision that conducted these "easy" and "natural" extensions (at leas partially) would be a more convincing response to the criticisms and lead to a paper that would clearly meet the bar of acceptance.

**Reviewer Concerns:**

The authors did not provide a revised version of the paper so any modifications were promised rather than delivered. As a result, this means that there is considerably less chance that concerns are adequately addressed. For example, the paper was not modified to better justify the assumptions or to provide extended theoretical results.

C1: The response clarified that the exploration of a formulation that takes cost into account is a valuable future research direction, but is not within the scope of the current work. This is a reasonable position.

C2. The response claims without clear evidence that the i.i.d. assumption be relaxed and that the  framework can be naturally extended to settings with gradual or continuous distribution shifts. If this were as straightforward as the response suggests, then it is unclear why the authors did not include it as an additional result in a revised version of the work. The response also argues that the piecewise stationary assumption is practically relevant, but does not provide any data analysis to show that this provides a reasonable match to practically observed data.

C3. The response argues that the contribution is theoretical so it is reasonable to provide very limited experiments that illustrate the concepts. While it is true that much less experimentation (and perhaps even none) is expected for a paper that focuses on making a theoretical contribution, when that theory is underpinned by and completely relies on strong assumptions, it is expected that there should be some analysis to demonstrate that the assumptions are at least reasonable approximations. Otherwise the analysis can become somewhat pointless.

**Reviewer Scores:**

Reviewer VQJi. UNLIKELY TO CHANGE BUT INADEQUATE REVIEW. The review provides 1.5 lines of weaknesses and less than a line of strengths. The review does not explain or clarify these viewpoints (e.g., "results are expected" - why? how?). As a result, the review has to be largely disregarded and it does not really matter whether the reviewer would change the score or not.

Reviewer TiTP. POSSIBLE INCREASE BUT UNLIKELY. The review identified only two relatively generic weaknesses focusing on related work and experiments to validate results. The author response offered to extend the related work to include the paper that the reviewer identified, but the criticism did not clearly specify other related work. The response did not provide any further empirical validation or support. Since the authors did not provide a revision and neither criticism was clearly addressed, it is likely that the score would remain unchanged.

Reviewer h3gc. UNLIKELY TO CHANGE. The core criticisms were not addressed in detail and there was not a revised paper. The response provided suggestions regarding how the theory could be extended but did not provide concrete results that demonstrated that these claimed extensions/relaxations were genuinely achievable.

Reviewer SH8c. UNLIKELY TO CHANGE. The core criticism from this reviewer focused on the assumptions, the lack of practical relevance, and the lack of validation. Since none of these were addressed in a revision, it is unlikely that the reviewer would change score.

Reviewer ZzpT: UNLIKELY TO CHANGE (Already a score of 8).

---

### Decision · Program_Chairs · 2026-01-26

Reject